# Memory-Driven Self-Improvement for Decision Making with Large Language Models

## Abstract

Large language models (LLMs) have emerged as effective action policies for sequential decision-making (SDM) tasks due to their extensive prior knowledge. However, this broad yet general knowledge is often insufficient for specific decision-making tasks with limited task-related data, making it challenging to efficiently adapt LLMs to specific SDM tasks. To address this challenge, we propose a memory-driven self-improvement framework that combines LLM general prior knowledge with a compact memory of domain-specific experiences. Memory retains past interactions and associated Q-values, thereby capturing decision-relevant knowledge that facilitates accurate value estimation and informs the LLM prior refinement. The refined LLM prior, in turn, generates higher-reward trajectories that further enrich memory, forming a natural self-improvement framework where memory and LLM prior mutually reinforce each other. Experiments show that our memory-driven approach significantly outperforms both traditional RL and LLM-based baselines, e.g., improving performance by over 40% on in-distribution tasks and over 75% when generalized to unseen tasks in ALFWorld.

## 1 Introduction

Sequential decision-making (SDM) has a wide range of real-world applications, including robotics Polydoros & Nalpantidis (2017); Brunke et al. (2022); Rana et al. (2023), autonomous driving Naranjo et al. (2005); Song et al. (2022), and human–AI interaction Granter et al. (2017); Li et al. (2019); McTear (2022). Natural language plays a crucial role in many SDM tasks, either in purely language-based settings Granter et al. (2017); Jannala (2025); Jin et al. (2024) or as a tool for understanding and describing the environment Ma et al. (2024a); Wang et al. (2025). Large language models (LLMs), with their broad prior knowledge, demonstrate strong zero-shot reasoning capabilities, making them promising candidates for action policies in such text-based SDM tasks. However, when deployed in specialized domains, their general knowledge is often insufficient for reliable decision-making Yan et al. (2025); Jannala (2025).

To adapt LLM action policies into the target domains, three main approaches have been explored. The first is prompt-based methods, which utilize human-crafted prompts Yao et al. (2022; 2024) or incorporate historical interactions Shinn et al. (2024); Christianos et al. (2023) to provide more task-specific information. However, these prompt-engineering methods heavily depend on the quality of the prompts and the reasoning capabilities of the LLMs. The second approach is fine-tuning, which includes supervised fine-tuning (SFT) and reinforcement learning fine-tuning (RLFT). SFT typically requires substantial high-quality decision-making data Zhou et al. (2024), while on-policy RLFT methods Carta et al. (2023); Tan et al. (2024) suffer from poor sample efficiency Abdolmaleki et al. (2018); Chen et al. (2023). The third pipeline performs RL with fixed LLM priors, using LLMs either to narrow the action search space Yan et al. (2025) or to design reward functions Kwon et al. (2023); Klissarov et al. (2023) that promote efficient exploration. However, these methods remain highly sensitive to the capability of the LLM priors in fulfilling such roles.

Considering these limitations, we propose a **memory-driven self-improvement** framework for text-based SDM. To combine the benefits of LLM general knowledge with task-specific interactions, a memory-driven action policy with LLM prior is designed, where the LLM prior generates action candidates, and memory-driven value estimation guides more precise action posterior selection. In practice, the framework forms a closed loop with two mutually reinforcing roles as illustrated in Figure 1: **Role 1**: **Memory-driven value estimation**, which converts informative interactions into

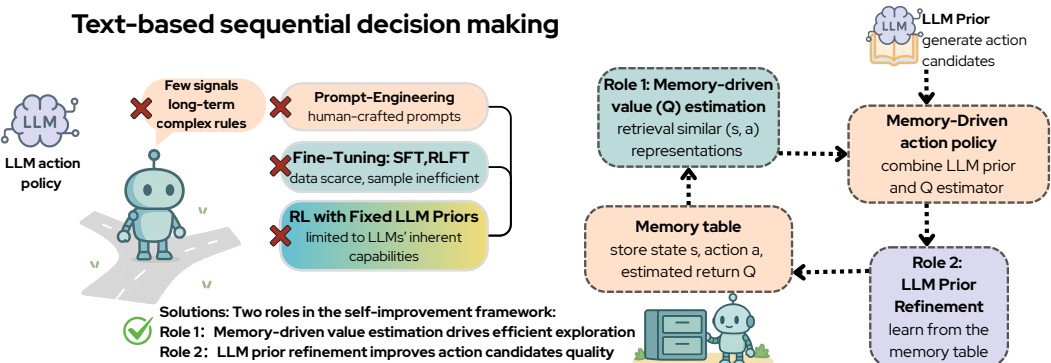

Figure 1: Motivation and overview of our memory-driven self-improvement framework for text-based SDM. Left: existing approaches (prompt-engineering, fine-tuning, and RL with LLM priors) struggle under sparse signals and domain-specific data. Right: Our framework introduces two complementary roles: (1) memory-driven value estimation, which enables efficient exploration, and (2) LLM prior refinement, which biases action generation toward high-quality candidates; together forming a self-improvement loop that resists scarce experience and enables efficient adaptation.

compact memory representations. By retrieving semantically similar past experiences, the model can make non-parametric value estimates for action candidates and enable informed exploration choices. **Role 2**: **Memory-driven LLM prior refinement**, which periodically updates the LLM's decision prior using historical state-action pairs and their Q-values stored in memory. This refinement biases the LLM prior toward generating high-quality actions, effectively narrowing the search space and improving the convergence rate. Overall, the informative memory table provides a reliable foundation for LLM prior refinement, while the refined LLM prior leads to higher-quality actions that further enrich the memory, thus naturally constructing a self-improvement framework. This mutual reinforcement enables scalable and efficient adaptation to target SDM tasks.

In summary, our main contributions are as follows:

1. We propose a **memory-driven self-improvement framework** for text-based SDMs and leverage the Expectation-Maximization (EM) to provide a unified formulation and practical implementation.

2. We introduce a **memory-driven value estimation** approach that utilizes LLMs' representation capabilities and retrieval techniques to achieve meaningful, non-parametric Q-value estimation.

3. We present a **memory-driven policy optimization** method that defines a powerful action policy as the combination of LLM priors and memory-based Q-estimation, and uses experiences stored in memory to refine the LLM prior.

4. Experimental results on ALFWorld and Overcooked demonstrate that memory-driven value estimation achieves superior sample efficiency, while LLM prior refinement proves crucial for further expanding the capabilities of LLM-based action policies.

## 2 PRELIMINARY

**Textual Markov Decision Processes.** A Markov Decision Process (MDP) is defined as $\langle \mathcal{S}, \mathcal{A}, \mathcal{P}, \mathcal{R}, \gamma \rangle$, where $\mathcal{S}$ is the state space, $\mathcal{A}$ is the action space, $\mathcal{P} : \mathcal{S} \times \mathcal{A} \to \mathcal{S}$ is the transition function, $\mathcal{R} : \mathcal{S} \times \mathcal{A} \to \mathbb{R}$ is the reward function, and $\gamma \in (0, 1)$ is the discount factor. Particularly, note that $r_t$ denotes the reward received at time step $t$. In this work, we focus on textual MDP, where both states and actions are represented in natural language, i.e., $\mathcal{S}, \mathcal{A} \subseteq \mathcal{V}^*$, with $\mathcal{V}$ denoting the vocabulary. Textual MDPs present unique challenges, as the state and action spaces can be combinatorially large, and policies must operate over inherently discrete, structured, and semantically rich representations of language.

**Q-learning for MDPs** A prominent paradigm for solving MDPs with discrete action spaces is Q-learning, where the Q-function $Q(s, a)$ is learned to rank and select actions. For each state–action

pair $(s, a)$, the Q-function, defined as $Q(s, a) = \mathbb{E}_\pi \left[ \sum_{i \geq t} \gamma^{i-t} r_i \mid s, a \right]$, represents the expected cumulative reward obtained by starting from $(s, a)$ and following policy $\pi$ thereafter. The DQN algorithm (Mnih, 2013) is a classic instance of Q-learning that maps state–action embeddings to scalar Q-values and trains the Q-network via temporal-difference (TD) learning. Blundell et al. (2016) introduces Episodic Control(EC), a memory-based Q-learning method that leverages retrieval techniques and informative state representations to enable non-parametric Q-value estimation.

**Control as Inference.** Control-as-Inference framework (Levine, 2018) formulates the solution of MDPs from a probabilistic inference perspective. An optimality random variable $\mathcal{O}$ is introduced, where $\mathcal{O} = 1$ indicates achieving a successful trajectory and $\mathcal{O} = 0$ otherwise. The objective is to maximize the likelihood of achieving a successful trajectory from each state $s$, which is identical to maximizing the evidence lower bound (ELBO):

$$\log p(\mathcal{O}{=}1|s) \geq \mathbb{E}_{q(a|s)}[\log p(\mathcal{O}{=}1|s,a)] - D_{\mathrm{KL}}(q(a|s)\|p(a|s)) \triangleq \mathrm{ELBO}, \quad (1)$$

where $p(a|s)$ is the prior action distribution, $q(a|s)$ is the variational distribution, and $p(\mathcal{O} = 1|s, a)$ is the likelihood that the trajectory will achieve optimality given the current state-action pair $(s, a)$. Abdolmaleki et al. (2018) assume that the optimality likelihood is proportional to Q-value: $p(\mathcal{O} = 1|s, a) \propto \exp(Q(s, a)/\tau)$. This probabilistic formulation offers both conceptual flexibility and methodological richness, while also naturally accommodating the integration of LLMs as sources of prior knowledge (Yan et al., 2025).

## 3 MEMORY-DRIVEN VALUE ESTIMATION WITH LLM PRIORS

In this section, we present a memory-driven approach to Q-value estimation that leverages LLM-based semantic representations. By explicitly exploiting LLM representations through retrieval techniques, our method builds upon the well-established episodic control (EC) (Blundell et al., 2016) to enable non-parametric Q-value estimation. Specifically, our method maintains a memory table $\mathcal{M}$ storing Q-values for visited state–action pairs, which is continuously updated during online exploration. At inference time, actions are selected through memory queries. The framework is thus defined by two core operations: **memory update** and **memory query**.

**Memory update.** A memory table $\mathcal{M}$ is constructed to store information about previously visited state-action pairs $(s, a)$. Specifically, it includes the natural language descriptions of $(s, a)$, the corresponding vectorized embeddings $f(s, a)$, and the associated Q-values $Q(s, a)$. The Q-values stored in memory $\mathcal{M}$ are then updated according to:

$$Q(s_t, a_t) \leftarrow \begin{cases} R_t, & \text{if } (s_t, a_t) \notin \mathcal{M}, \\ \max\{Q(s_t, a_t), R_t\}, & \text{otherwise,} \end{cases} \quad (2)$$

where $R_t = \sum_{i \geq t} \gamma^{i-t} r_i$ represents a Monte Carlo estimate of the cumulative discounted reward (i.e., the return-to-go).

---

**Algorithm 1** Memory-Driven Q-learning (Mem-Q)

**Input:** Embedding function $f_{\mathrm{LLM}}$, empty memory table $\mathcal{M} = \emptyset$.
**Output:** Updated memory table $\mathcal{M}$.
1: **for** each episode **do**
2:    **for** $t = 1, 2, 3, \ldots, T$ **do**
3:       Obtain all feasible actions $\mathcal{A}_{s_t}$ for current $s_t$.t
4:       Estimate the return $\widehat{Q}$ for each $a \in \mathcal{A}_{s_t}$ via Eq. 3.
5:       Select action $a_t$ using the $\epsilon$-greedy strategy.
6:       Execute action $a_t$, receive reward $r_{t+1}$, and observe the next state $s_{t+1}$.
7:    **end for**
8:    **for** $t = T, T-1, \ldots, 1$ **do**
9:       Write and update memory table $\mathcal{M}$ via Eq. 2.
10:    **end for**
11: **end for**

---

**Memory query.** Given the memory $\mathcal{M}$ and the current state $s_t$, the policy is then determined using the kernel-based Q-value estimator $\widehat{Q}$, which is defined as

$$\widehat{Q}(s_t, a) = \sum_{i \in \mathcal{N}_M(f(s_t, a))} w_i Q(s^{(i)}, a^{(i)}), \quad w_i = \frac{k(h, h_i)}{\sum_{j \in \mathcal{N}_M} k(h, h_j)}, h = f(s_t, a), h_i = f(s^{(i)}, a^{(i)}), \quad (3)$$

where $\mathcal{N}_M(f(s_t, a))$ denotes the $M$ nearest neighbors of $f(s_t, a)$, and the inverse distance kernel is used with $k(h, h_j) = \frac{1}{\|h - h_j\|_2^2 + \delta}$, which measures similarity in the embedding space. With the estimated Q-values $\widehat{Q}$, action selection can be carried out using the $\epsilon$-greedy strategy, analogous to the approach in DQN (Mnih, 2013).

**Vectorized Memory Query with LLM Priors.** While conceptually simple, the effectiveness of the proposed memory-driven Q-learning critically depends on the choice of embedding function $f$. Inspired by the recent success of Retrieval Augmented Generation (RAG) (Wiratunga et al., 2024), which demonstrates the ability of LLMs to yield semantically rich vectorized representations, we leverage LLM-based embeddings for $(s, a)$ pairs to enhance retrieval quality and improve value estimation. Specifically, we

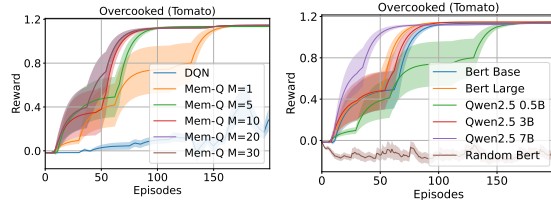

Figure 2: Results of memory-drive Q-learning on Overcooked. Left: effect of the number of retrieved $(s, a)$ pairs for value estimation; Right: effect of different LLMs on representations.

adopt the encoder of the LLM to be the embedding function $f_{\mathrm{LLM}}$. The overall memory-driven decision-making procedure with LLM embedding function is summarized in Algorithm 1, referred as Memory-Driven Q-learning(Mem-Q). It is noteworthy that Blundell et al. (2016) proposes a similar approach, episodic control, but it operates based solely on state similarity. This requires maintaining $|\mathcal{A}|$ separate memory tables, one for each feasible action $a \in \mathcal{A}$, thereby restricting the method to small and enumerable action spaces while ignoring semantic similarity between actions.

In Figure 2, we evaluate the Mem-Q in Overcooked (Tan et al., 2024), a textual decision-making task. It shows that our Mem-Q significantly outperforms DQN. The ablation study on the retrieval size $M$, introduced in Eq. 3, shows that incorporating more similar $(s, a)$ embeddings in the kernel-based Q estimation leads to faster convergence, which may be attributed to the more accurate value estimation using more meaningful representations (Han et al., 2023). Furthermore, larger LLMs, which capture richer semantic structures, consistently yield stronger performance.

## 4 MEMORY-DRIVEN POLICY OPTIMIZATION WITH LLM PRIORS

The success of memory-driven Q-learning highlights a key insight: combining the capabilities of LLMs with experience storage can substantially enhance the efficiency of RL algorithms. Motivated by this, we propose a **memory-driven action policy** that uses the LLM prior to narrow the action search space, and leverage memory to further refine the LLM prior with domain-specific knowledge, thereby improving sample efficiency.

### 4.1 MEMORY-DRIVEN POLICY WITH LLM PRIORS

A straightforward approach to incorporating LLMs into the action policy is through a probabilistic inference framework (Li et al., 2024), wherein the policy is cast as the posterior distribution:

$$p_\theta(a|s, \mathcal{O} = 1) \propto p(\mathcal{O} = 1|s, a)p_{\mathrm{LLM}_\theta}(a|s), \tag{4}$$

where $p_{\mathrm{LLM}_\theta}(a|s)$ is the LLM prior parametrized by $\theta$ and $p(\mathcal{O} = 1|s, a)$ denotes the likelihood of optimality[1], which can be estimated using Q-value, as described in Eq 3. Therefore, the memory-driven policy can be conducted using self-normalized importance sampling as follows:

- Sample $K$ candidate actions from the LLM prior: $\mathcal{C}^K(s) = \{a_1, \ldots, a_K\}, a_k \sim p_{\mathrm{LLM}_\theta}(\cdot|s)$.

- For each action candidate $a_k$, approximate the optimality likelihood using the kernal-based Q-value estimator $\widehat{Q}(s, a_i)$ according to Eq 3.

- Select an action via: $a \sim \mathrm{Multinormial}\left(\exp(\widehat{Q}(s, a_1)/\tau)/\sum_k \exp(\widehat{Q}(s, a_k)/\tau)\right)$.

Although conceptually straightforward, this method is sensitive to the specification of the LLM prior $p_\theta$. A poorly aligned prior may bias the candidate set toward suboptimal actions, thereby degrading policy performance. To overcome this limitation, the following section introduces a principled optimization procedure based on the Expectation–Maximization framework, which refines the memory-driven policy by adaptively optimizing the underlying LLM priors.

---

[1]Notably, the likelihood exhibits no explicit parametric dependence, as the proposed memory-driven Q-learning framework is inherently non-parametric.

---

**Algorithm 2** Memory-Driven Expectation Maximization (Mem-EM) with LLM Prior Refinement

---

**Input:** LLM action prior $p_{\mathrm{LLM}_\theta}$, memory table $\mathcal{M} = \emptyset$
**Output:** Refined LLM prior $p_{\mathrm{LLM}_\theta}$ and memory table $\mathcal{M}$
1: **for** episode $i = 1$ to $N$ **do**
2:     **for** step $t = 1, 2, \ldots, T$ **do**
3:         Sample action candidates from LLM prior $\mathcal{C}^K(s_t) \sim p_{\mathrm{LLM}_\theta}(\cdot|s_t)$.
4:         Estimate Q-values $\widehat{Q}(s_t, a)$ for $a \in \mathcal{C}^K(s_t)$ via Eq. 3.
5:         Select action $a \sim \mathrm{Multinormial}(\exp(\widehat{Q}(s, a_1)/\tau)/\sum_k \exp(\widehat{Q}(s, a_k)/\tau))$.
6:         Execute $a_t$, observe reward $r_{t+1}$ and next state $s_{t+1}$
7:     **end for**
8:     **for** step $t = T$ down to 1 **do**
9:         Write and update memory table $\mathcal{M}$ via Eq. 2.
10:     **end for**
11:     **if** $i$ reaches update interval **then**
12:         Refine LLM prior $p_{\mathrm{LLM}_\theta}$ using stored $(s, a)$ pairs from $\mathcal{M}$ via Eq. 7.
13:     **end if**
14: **end for**

---

**Note:** Orange text highlights LLM-guided exploration steps that differ from the Mem-Q in Algorithm 1; green text indicates prior refinement operations.

---

## 4.2 Optimizing Memory-Driven Policy with Expectation Maximization

To optimize the memory-driven policy with the LLM prior and the likelihood estimation involved, we adopt the Expectation–Maximization (EM) algorithm. Starting from an arbitrary initialization $\theta_0$, the EM procedure performs the following iterative update:

$$\theta_{k+1} = \arg\max_\theta \mathbb{E}_{p_{\theta_k}(a, s|\mathcal{O}=1)} \left[ \log p(\mathcal{O} = 1|s, a) + \log p_{\mathrm{LLM}_\theta}(a|s) \right]. \tag{5}$$

By construction, the EM update can be interpreted as maximizing the ELBO in Eq. 1. In practice, however, the expectation in Eq. 5 is analytically intractable. To address this challenge, we employ an on-policy E-step for likelihood expectation approximation and a memory-driven off-policy E-step for prior expectation approximation, followed by maximization based on these tractable estimates.

**Expectation Step.** In the E-step, a simple yet tractable approach for the expectation approximation is the Monte Carlo (MC) estimation. Concretely, for the current state $s$, we draw an action from the posterior $a \sim p_\theta(a|s, \mathcal{O} = 1)$, which can be approximated using importance sampling introduced in Sec. 4.1. For the likelihood expectation term in Eq. 5, the likelihood is assumed to be proportional to the Q value, and one can directly apply the kernel-based Q value estimation in Eq. 3.

However, for the LLM prior expectation term, such an "on-policy" approach is inefficient, since LLMs typically involve a very large number of parameters, and a limited set of MC samples is insufficient to provide reliable gradient estimates for subsequent M-step. Consequently, we consider importance sampling approximation using examples stored in the memory table to explore and exploit the posterior distribution effectively, thereby yielding a robust "off-policy" estimate of the expectation. Specifically, we employ self-normalized importance sampling (SNIS) to estimate the LLM prior expectation as follows:

$$\mathbb{E}_{p_\theta(a, s|\mathcal{O}=1)} \left[ \log p_{\mathrm{LLM}_\theta}(a|s) \right] = \mathbb{E}_{q(s,a)} \left[ \frac{p_\theta(a, s|\mathcal{O} = 1)}{q(s, a)} \log p_{\mathrm{LLM}_\theta}(a|s) \right]$$

$$\approx \sum_i \frac{w(s^{(i)}, a^{(i)})}{\sum_j w(s^{(j)}, a^{(j)})} \log p_{\mathrm{LLM}_\theta}(a^{(j)}|s^{(j)}), \ (s, a) \sim q(s, a)$$

where $w(s, a) = \frac{p(\mathcal{O}=1|a, s) p_{\mathrm{LLM}_\theta}(a, s)}{q(s, a)}$ denotes the importance weight. While the proposal $q(s, a)$ offers substantial flexibility, the statistical efficiency of SNIS is highly sensitive to its choice: suboptimal proposals yield high-variance importance weights, thereby impeding effective state–action exploration. In theory, the optimal proposal, which leads to zero variance, is $q(s, a) \propto p(\mathcal{O} = 1|a, s) p_{\mathrm{LLM}_\theta}(a, s)$. While intractable, empirically, the memory table $\mathcal{M}$ provides a practical basis

for designing the proposal distribution $q(s, a)$, as it stores state-action pairs accumulated from previous posterior samples. We thus adopt a uniform distribution over the memory table $\mathcal{M}$ as the proposal, providing an empirical approximation to the posterior. Specifically, the importance weight can be approximated as:

$$w(s, a) = \frac{p(\mathcal{O} = 1|a, s)p_{\text{LLM}_\theta}(a, s)}{q(s, a)} \approx \frac{p(\mathcal{O} = 1|a, s)q(a, s)}{q(s, a)} \propto \exp(Q(s, a)/\tau) \qquad (6)$$

where we further use the proposal $q(s, a)$, which is the emperical distribution of the memory table $\mathcal{M}$, to approximate the joint LLM prior $p_{\text{LLM}_\theta}(s, a)$. Although heuristic, this approximation is empirically found to stabilize training. Intuitively, at iteration $k$, the optimal LLM prior coincides with the posterior from the previous step, $p_{\theta_k}(a|s, \mathcal{O} = 1)$, as suggested by Eq. 5. Hence, the intractable LLM prior can be approximated by the empirical distribution $q(s, a)$, functioning as a moving average of the posterior. This approach eliminates the need for repeated LLM queries to estimate the prior, thereby reducing computational cost and improving training efficiency.

**Maximization Step.** In the M-step, we maximize the expectation in Eq. 5, which involves the optimality likelihood $p(\mathcal{O} = 1 \mid s, a)$ and the LLM prior $p_{\text{LLM}_\theta}$. To maximize the expectation of the non-parametric likelihood, we update the memory according to Eq. 2 using the sample drawn from the posterior. For optimizing the LLM prior, building on the memory-driven expectation estimation, we apply the following memory-based reweighted training objective:

$$\theta_{k+1} = \arg\min_\theta \sum_{(s,a)\sim\mathcal{M}} \left[ \frac{\exp(Q(s, a)/\tau)}{\sum_{(s',a')\sim\mathcal{M}} \exp(Q(s', a')/\tau)} \log p_{\text{LLM}_\theta}(a|s) \right]. \qquad (7)$$

We summarize the detailed training procedure in Algorithm 2. It is noteworthy that memory-driven Q estimation and memory-driven policy optimization interact in a bootstrapping manner. Specifically, the memory-driven Q estimation module continuously refines the non-parametric estimate of the optimality likelihood, which in turn provides more accurate feedback for updating the LLM prior. Conversely, the improved LLM prior constrains the action search space toward high-quality candidates, thereby accelerating convergence and improving the sample efficiency of memory-driven Q estimation. This closed-loop interaction establishes a self-improvement cycle, where the two components iteratively enhance one another, leading to progressively stronger policies.

Although both from the probabilistic inference aspect to solve decision-making tasks, our memory-driven self-improvement framework is highly different from previous control as inference works. Specifically, traditional control-as-inference methods like MPO (Abdolmaleki et al., 2018) rely on sampling from and estimating the action policy over the entire action space, limiting their adaptability to complex scenarios such as ALFWorld(Shridhar et al., 2020) with huge and state-varying action spaces. In contrast, our method leverages the LLM prior to generating $K$ action candidates and then uses memory-driven Q-estimation to select the best action. This mechanism enables our method to adapt to complex unstructured action space settings. Furthermore, this Q-value guided action policy effectively narrows down the executable exploration space, enabling efficient exploration.

## 5 EXPERIMENTS

In this section, we design experiments to validate the effectiveness of our memory-driven self-improvement framework.

### 5.1 ENVIRONMENTS

We consider two textual decision-making tasks:
**Overcooked** The textual Overcooked environment (Tan et al., 2024) benchmarks the ability of taking a sequence of actions to deliver dishes. We consider two Overcooked tasks: Overcooked(Tomato), which requires delivering a dish of chopped tomato, and Overcooked(Salad), which requires delivering a salad containing chopped tomato and lettuce. The feasible actions vary with state changes, and the maximum number of feasible actions for one state is 8. Besides the reward of 1 for successfully delivering a dish, this environment also provides the following reward shaping: +0.2 for correctly chopping an ingredient, +1 terminal reward for successfully delivering the correct dish, -0.1 for delivering any incorrect item, and -0.001 for each time step.

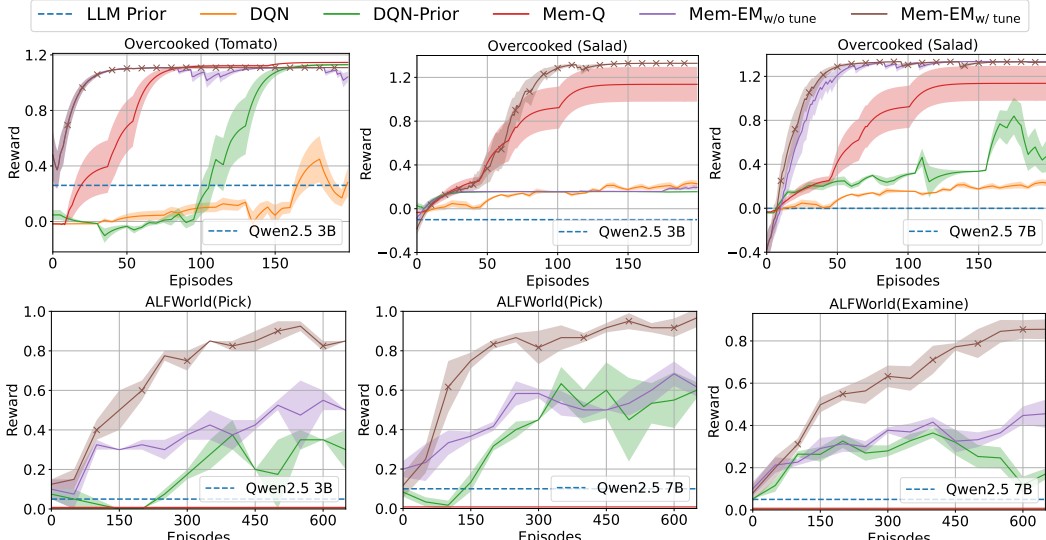

Figure 3: Results of comparison with baselines. We plot the mean and standard error of the cumulative reward. The dashed line represents directly prompting the LLM prior to generating actions given state information, with the corresponding LLM version specified. The '×' markers for Mem-EM_{w/ tune} indicate the time steps when the LLM prior is fine-tuned.

**ALFWorld** (Shridhar et al., 2020) is a popular and complex decision-making benchmark for navigating and completing tasks within rooms. Both observations and actions are textual, and the action space consists of high-level plans such as "go to a room" that can be understood using LLMs' prior knowledge. The action space is finite but large, with varying feasible action sets for each state, and the maximum possible admissible action space for one step reaches up to 50, making exploration from scratch challenging. This benchmark contains thousands of subtasks, making it convenient for testing generalization performance on unseen tasks. We consider two classes of ALFWorld tasks: ALFWorld(pick) and ALFWorld(examine). There are no auxiliary rewards except for a reward of 1.0 for reaching the final goal.

## 5.2 BASELINES

We compare against the following baselines:

**LLM prior**: We first assess the ability of LLMs to solve decision-making tasks in a zero-shot manner, without relying on deliberately designed prompts.

**DQN-based methods**: We compare against **DQN** and its variant **DQN-prior** Yan et al. (2025), which performs deep Q-learning within a narrowed sub-action space generated by LLM priors.

Our memory-driven algorithms: **Mem-Q** (Algorithm 1) leverages language models' embedding capabilities and the kernel-based value estimation to perform non-parametric Q-learning.

**Mem-EM** (Algorithm 2) formulates decision-making as an EM procedure that integrates memory-driven Q-value estimation with LLM prior refinement. We consider two variants: **Mem-EM_{w/ tune}** fine-tunes the LLM prior using examples from the memory table at regular intervals, and **Mem-EM_{w/o tune}** uses the fixed LLM prior without fine-tuning.

**Experimental Settings** We use Qwen2.5-3B (Team, 2024) or Qwen2.5-7B as the LLM prior for our main experiments. The retrieval size $M$ is set to 20, and the number of action candidates $K$ is set to 5 or 10. The fixed BERT-base (Devlin et al., 2019) model is used to obtain semantic representations of state–action pairs. Detailed hyperparameter settings are provided in the Appendix 10.

## 5.3 RESULTS

**Comparison with Baselines.** The comparison results of baselines are shown in Figure 3. These results demonstrate that Mem-EM_{w/ tune} outperforms all other baselines, remarkably achieving over 40% improvement on complex ALFWorld environments. We observe that memory-based Mem-Q significantly outperforms DQN on Overcooked, and memory-based Mem-EM_{w/o tune} achieves comparable or better performance than DQN-prior, demonstrating that the memory-driven approaches

Table 1: Results on the generalization ability of ALFWorld(Pick). All trainable models are trained with $K = 5$ action candidates, and their performance is evaluated using different values of $K$.

| Baseline | Unseen Tasks | | | | Seen Tasks | | | |
|---|---|---|---|---|---|---|---|---|
| | K=5 | K=10 | K=15 | K=20 | K=5 | K=10 | K=15 | K=20 |
| LLM | 0.19 | | | | 0.1 | | | |
| LLM + $Q_{\text{DQN-Prior}}$ | 0.16 | 0.19 | 0.14 | **0.22** | 0.6 | **0.70** | **0.70** | 0.65 |
| LLM + $Q_{\text{Mem-EM}_{\text{w/tune}}}$ | 0.22 | 0.27 | **0.35** | 0.22 | 0.65 | 0.80 | **0.85** | 0.65 |
| $\text{LLM}_{\text{Mem-EM}_{\text{w/tune}}}$ | 0.59 | | | | 0.85 | | | |
| $\text{LLM}_{\text{Mem-EM}_{\text{w/tune}}} + Q_{\text{DQN-Prior}}$ | **0.59** | 0.54 | 0.46 | 0.46 | **0.90** | 0.85 | **0.90** | 0.75 |
| $\text{LLM}_{\text{Mem-EM}_{\text{w/tune}}} + Q_{\text{Mem-EM}_{\text{w/tune}}}$ | **0.81** | 0.65 | 0.62 | 0.68 | 0.95 | 0.95 | **1.00** | 0.95 |

provide satisfactory value estimation and improve sample efficiency. However, for the complex ALFWorld, where both state and action spaces are extremely large, neither DQN nor Mem-Q alone can solve the task, highlighting the necessity of incorporating LLM priors.

By leveraging LLM priors to generate valuable action candidates, the baselines DQN-Prior and Mem-EM$_{\text{w/o tune}}$ consistently outperform their vanilla counterparts (DQN and Mem-Q) that explore the full original action space. Nevertheless, since pretrained LLMs lack task-specific knowledge, fixed LLM priors inherently limit performance and may even degrade it. For example, in the Overcooked (Salad) task with Qwen2.5-3B and $K = 10$ candidate actions, incorporating the prior actually degrades performance. This suggests that Qwen2.5-3B cannot construct a reliable sub-action space that consistently includes the optimal action for each state, due to its insufficient domain knowledge and decision-making capability.

Importantly, results show that Mem-EM$_{\text{w/ tune}}$ substantially outperforms Mem-EM$_{\text{w/o tune}}$ on ALFWorld and Overcooked (Salad) with the 3B model. This demonstrates that our memory-driven policy optimization effectively integrates domain-specific knowledge into the LLM prior, thereby improving its decision-making ability. Furthermore, as illustrated in the ALFWorld experiments, the EM-based framework Mem-EM$_{\text{w/ tune}}$ requires only six time LLM tuning throughout training, with LoRA Hu et al. (2021) used for parameter-efficient fine-tuning. This keeps the computational overhead tolerable while yielding significant performance improvements.

**Generalization Ability.** The generalization ability evaluation results on ALFWorld(Pick) are shown in Table 1. We evaluate the generalization ability of the finetuned LLM policy and the Q-value estimators including the Q-network in and the memory-driven Q-estimator defined in Eq. 3. In general, we consider the components below and their combinations: the pretrained Qwen2.5-7B, denoted as **LLM**; the Q network trained with DQN-Prior, denoted as $\mathbf{Q_{\text{DQN-Prior}}}$; the fine-tuned LLM following Mem-EM$_{\text{w/ tune}}$, denoted as $\mathbf{LLM_{\text{Mem-EM}_{\text{w/ tune}}}}$; and the Q estimator of Mem-EM$_{\text{w/ tune}}$, denoted as $\mathbf{Q_{\text{Mem-EM}_{\text{w/ tune}}}}$. For example, the Mem-EM$_{\text{w/tune}}$ generates a refined LLM and a memory table of $(s, a, Q)$ tuples. The LLM$_{\text{Mem-EM}_{\text{w/tune}}}$ uses the refined LLM to directly generate a single action for execution. The LLM$_{\text{Mem-EM}_{\text{w/tune}}} + Q_{\text{Mem-EM}_{\text{w/tune}}}$ uses the refined LLM to generate $K$ action candidates and then selects one action based on value estimation following the memory table.

The results show that while all methods perform well on seen tasks, the Q-estimators achieve only modest improvements over the pretrained LLM on unseen tasks. This exposes the generalization limitations of Q-estimators, which are constrained by the BERT-base representations despite being effective on seen tasks. In contrast, LLM fine-tuning shows superior generalization ability on unseen tasks, demonstrating the necessity of LLM prior refinement. Combining the fine-tuned LLM with the memory-based Q-estimator further improves performance, achieving a over 75% performance gain than the pretrained LLM.

On unseen tasks, the phenomenon of the better performance of LLM$_{\text{Mem-EM}_{\text{w/ tune}}} + \mathbf{Q_{\text{Mem-EM}_{\text{w/ tune}}}}$ with $K = 5$ (following the training setting) compared to larger $K$ may stem from the introduction of low-quality, noisy samples from the tail distribution of the refined LLM prior. Specifically, the refined LLM has distilled domain-specific decision-making knowledge and is capable of generating high-quality actions when $K = 5$. However, when $K$ is increased, less relevant or noisy action candidates are more likely to be sampled from the refined LLM prior's low-probability tail. This inclusion of noisy actions, combined with the inherent generalization limitations of the value estimation model, degrades the final policy performance. Overall, these results indicate that K=5 balances sufficient exploration coverage with high-quality action candidates.

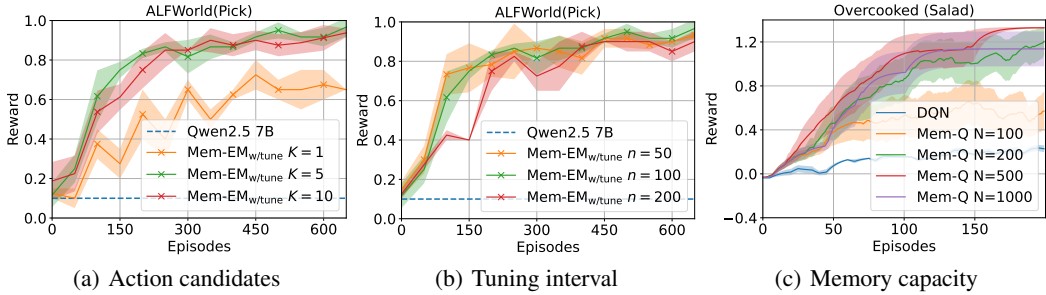

(a) Action candidates      (b) Tuning interval      (c) Memory capacity

Figure 4: Ablation study results. (a) Effect of the number of action candidates $K$ generated by the LLM. The '$\times$' markers indicate the time steps when the LLM-prior is updated. (b) Impact of the LLM fine-tuning interval, where $n$ denotes that the LLM policy is fine-tuned every $n$ episodes. (c) Influence of the memory table capacity $N$, where at most $N$ $(s, a)$ pairs are stored, with the least-recently-used (LRU) strategy applied for replacement.

## 5.4 ABLATION STUDIES

The ablation study results are shown in Figure 4, where we analyze the following aspects:

**Effect of the number of action candidates.** Figure 4(a) illustrates the impact of the number of action candidates $K$ for Mem-EM$_{\text{w/ tune}}$. Our method with $K = 5$ or 10 significantly outperforms the setting with $K = 1$, indicating that it benefits from both (1) LLM fine-tuning that incorporates domain-specific knowledge stored in the memory table, and (2) Q-value guided posterior action sampling. It is worth noting that the case of Mem-EM$_{\text{w/ tune}}$ with $K = 1$ resembles an actor–critic RL setting, where the action policy samples trajectories and the critic model is trained with environment returns to guide the policy learning. These results highlight the superiority of our EM-based training framework over traditional actor–critic RL framework. This superiority is achieved because, in our approach, the policy is treated as a prior, while sampling is performed from the action posterior and guided by memory-driven value estimation, jointly resulting in higher sample efficiency.

**Effect of the LLM prior update interval.** Fig. 4(b) examines the effect of the fine-tuning interval $n$, where the LLM prior is updated every $n$ episodes. Results show that Mem-EM$_{\text{w/ tune}}$ is robust to the update interval and does not require frequent fine-tuning to achieve strong performance.

**Effect of memory capacity.** Figure 10 shows the effect of memory table capacity $N$ on Overcooked (Salad). In this environment, the total number of possible $(s, a)$ pairs is approximately 1000. Remarkably, Mem-Q with only $N = 100$ entries already outperforms DQN, and configurations with $N = 200$ or 500 achieve performance comparable to $N = 1000$, which stores all possible pairs. This demonstrates that, by simply using a least-recently-used (LRU) replacement strategy to retain crucial $(s, a)$ pairs, memory-driven Q estimation remains robust to memory capacity.

## 6 RELATED WORK

**LLM Priors in Decision-Making** Recent works leverage LLMs to enhance sequential decision-making (SDM) in three main ways: action generation, value estimation, and reward function design. First, LLMs can act as the action policy, generating satisfactory actions either through deliberate prompting Yao et al. (2022); Shinn et al. (2024) or RL-based fine-tuning Carta et al. (2023); Tan et al. (2024). Second, LLMs can serve as the value function to guide the search. Examples include reasoning-path search Wang et al. (2022); Yao et al. (2023) and Monte Carlo Tree Search guided by LLM evaluations Hao et al.; Wan et al. (2024). In addition, LLMs can be fine-tuned to act as process or outcome reward models with detailed explanations Lightman et al. (2023); McAleese et al. (2024); Wang et al. (2024b). Third, LLMs are used to generate reward signals for RL, either directly by prompting with historical interactions Kwon et al. (2023) or by producing executable reward code for continuous-control tasks Yu et al. (2023); Ma et al. (2024b).

**Memory-based Decision-Making** Episodic Control (EC) Blundell et al. (2016) and its extensions Pritzel et al. (2017); Li et al. (2023) represent a classic family of memory-based methods. These approaches maintain $|\mathcal{A}|$ separate memory tables, one for each feasible action, and apply kernel-based estimation over state representations. Yet, EC relies solely on state similarity and ignores semantic relationships among actions. Off-policy methods such as DQN Mnih (2013) and SAC Haarnoja et al. (2018) can also be viewed as memory-based RL, with the replay buffer serving

as memory. Yan et al. (2024) further combine DQN with LLM embeddings, but their approach compresses stored experiences into Q-networks that map high-dimensional embeddings to scalar values, potentially discarding semantic information and limiting sample efficiency. More recently, memory-based methods have been integrated with LLMs under the retrieval-augmented generation (RAG) paradigm Zhou et al. (2025); Wang et al. (2024a). These approaches retrieve relevant cases to enhance LLM outputs via in-context learning Han et al. (2023), but they primarily focus on one-step tasks such as question answering Wiratunga et al. (2024) or high-level planning Zhou et al. (2025). By contrast, our work applies retrieval techniques to sequential decision-making, and explicitly incorporates domain-specific memory to refine LLM-based policies.

## 7 CONCLUSIONS

In this work, we propose a memory-driven self-improvement framework for decision-making tasks. The framework consists of two mutually reinforcing components: memory-driven value estimation and memory-driven LLM prior refinement. The memory-driven value estimation approach maintains historical interactions and their Q-values, performing non-parametric value estimation through retrieval of similar representations. Building on the EM formulation, we design a practical and stable memory-driven LLM prior refinement algorithm, which adapts task-specific knowledge into the LLM prior by learning from the memory table. The explicit use of memory in these two components encourages efficient exploration. Experimental results show that our EM-based self-improvement framework delivers substantial performance gains while avoiding extensive fine-tuning. In this work, we focus on text-based decision-making with discrete but enumerable action spaces. For future research, we plan to extend our framework to handle scenarios with free-form or infinite action spaces and to incorporate vision–language models, thereby enabling broader applications.

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

# 8 ADDITIONAL EXPLAINATION ON MEMORY-DRIVEN VALUE ESTIMATION

In the main paper, we presented memory-driven Q-learning estimation without training. Here, we explore an alternative approach that involves tuning the embedding function. As described above, we use the BERT-base model as the embedding function $f_\theta$, which can be further fine-tuned by minimizing the distance between the predicted Q-value and the Monte Carlo estimate:

$$\ell = -(\hat{Q}_\theta(s_t, a) - y_t)^2,$$

where $y_t = \sum_{i=t}^{T} \gamma^{i-t} r_t$. The results of fine-tuning the BERT embedding model are shown in Figure 5. We observe that our model, both with and without fine-tuning, outperforms DQN-Prior. All three baselines use the LLM prior to narrow the action search space, and fine-tuning versus keeping the BERT embedding model fixed yields similar performance.

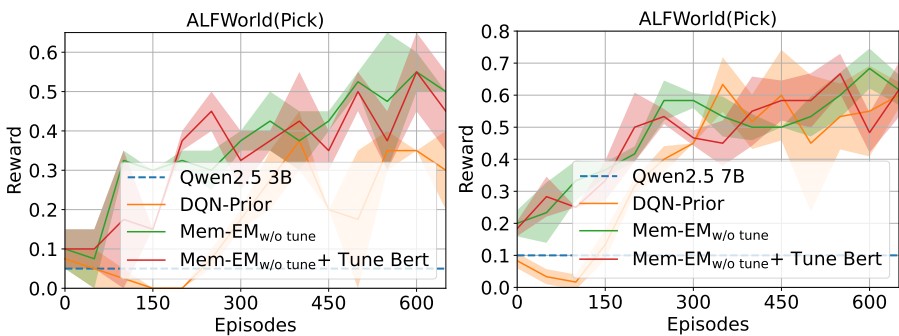

Figure 5: Ablation study on finetuning the embedding model bert.

# 9 ADDITIONAL EXPLANATION OF PROBABILISTIC INFERENCE

In this section, we briefly review the Expectation–Maximization (EM) algorithm. We start by introducing the evidence lower bound (ELBO) for the log-likelihood:

$$\log p_\theta(\mathcal{O} = 1|s) = \sum_a \log p_\theta(\mathcal{O} = 1|s, a) p_\theta(a|s) \tag{8}$$

$$= \sum_a \log \left[ q_\phi(a|s) \frac{p_\theta(\mathcal{O} = 1|s, a) p_\theta(a|s)}{q_\phi(a|s)} \right] \tag{9}$$

$$\geq \mathbb{E}_{q_\phi(a|s)}[\log p_\theta(\mathcal{O} = 1|s, a)] - D_{\text{KL}}(q_\phi(a|s) \| p_\theta(a|s)) \triangleq \text{ELBO}(\phi, \theta) \tag{10}$$

In variational inference, maximizing the likelihood can be reformulated as the following two-step optimization:

$$\phi_{k+1} \leftarrow \arg\max_\phi \text{ELBO}(\phi, \theta_k) \tag{11}$$

$$\theta_{k+1} \leftarrow \arg\max_\theta \text{ELBO}(\phi_{k+1}, \theta). \tag{12}$$

It can be shown that when $q_{\phi_{k+1}} = p_{\theta_k}(a|s, \mathcal{O} = 1)$, the ELBO attains its maximum value, which coincides with the true log-likelihood:

$$\mathbb{E}_{p_\theta(a|s, \mathcal{O}=1)}[\log p_\theta(\mathcal{O} = 1|s, a)] - D_{\text{KL}}(p_\theta(a|s, \mathcal{O} = 1) \| p_\theta(a|s)) \tag{13}$$

$$= \mathbb{E}_{p_\theta(a|s, \mathcal{O}=1)} \log \left[ \frac{p_\theta(a, \mathcal{O} = 1|s)}{p_\theta(a|s, \mathcal{O} = 1)} \right] \tag{14}$$

$$= \mathbb{E}_{p_\theta(a|s, \mathcal{O}=1)} \log \left[ \frac{p_\theta(a|s, \mathcal{O} = 1) p_\theta(\mathcal{O} = 1|s)}{p_\theta(a|s, \mathcal{O} = 1)} \right] \tag{15}$$

$$= \mathbb{E}_{p_\theta(a|s, \mathcal{O}=1)}[\log p_\theta(\mathcal{O} = 1|s)] \tag{16}$$

$$= \log p_\theta(\mathcal{O} = 1|s). \tag{17}$$

Thus, EM maximizes the following objective:

$$\theta_{k+1} = \arg\max_{\theta} \mathbb{E}_{p_{\theta_k}(a|s,\mathcal{O}=1)}[\log p_\theta(\mathcal{O}=1|s,a)] - D_{\text{KL}}(p_{\theta_k}(a|s,\mathcal{O}=1)\|p_\theta(a|s)), \quad (18)$$

which is equivalent to

$$\theta_{k+1} = \arg\max_{\theta} \mathbb{E}_{p_{\theta_k}(a,s|\mathcal{O}=1)}\left[\log p_\theta(\mathcal{O}=1|s,a) + \log p_\theta(a|s)\right]. \quad (19)$$

Although this optimization is generally intractable, in the E-step one can approximate the expectation using Monte Carlo estimation, followed by the M-step to update the parameter $\theta$. Finally, since the ELBO lower bounds the log-likelihood, each EM iteration guarantees monotonic improvement $\log p_{\theta_{k+1}}(\mathcal{O}=1|s) \geq \log p_{\theta_k}(\mathcal{O}=1|s)$.

## 10 DETAILED EXPERIMENTAL SETTINGS

### 10.1 ENVIRONMENTS

Examples of observations and admissible actions for ALFWorld and Overcooked are shown below:

---

**For ALFWorld(Pick)**

**Observation:** Task: Your task is to: put some knife on sidetable. Current observation:You open the drawer 2. The drawer 2 is open. In it, you see nothing..
**Admissible actions** You are allowed to take the following actions: close drawer 2, examine drawer 2, go to cabinet 1, go to cabinet 2, go to cabinet 3, go to cabinet 4, go to coffeemachine 1, go to countertop 1, go to drawer 1, go to drawer 3, go to drawer 4, go to drawer 5, go to drawer 6, go to drawer 7, go to drawer 8, go to fridge 1, go to garbagecan 1, go to microwave 1, go to sidetable 1, go to sinkbasin 1, go to stoveburner 1, go to stoveburner 2, go to stoveburner 3, go to stoveburner 4, go to toaster 1, inventory, look.

---

**For Overcooked(Salad)**

**Observation:** There are two fixed cutting boards in the room. You notice a tomato and an onion on the different tables. Currently you are carrying an unchopped lettuce in hand. To serve the dish of a bowl only containing chopped tomato and lettuce, what action should you take next?
**Admissible actions** You are allowed to take the following actions: 'pick up the tomato', 'pick up the lettuce', 'pick up the onion', 'take the empty bowl', 'put the lettuce on the first cutting board', 'put the lettuce on the second cutting board', 'serve the dish', 'chop nothing

---

### 10.2 LLM PRIOR IMPLEMENTATION

Following Yan et al. (2024), we use the LLM prior to generate $K$ action candidates by sampling $K$ free-form outputs in parallel from the LLM, given the current state and all admissible actions. These outputs are then mapped to executable actions via a simple rule-based projection $\mathcal{P}$. Mathematically, this can be described as: $a \sim p_{\text{LLM}}(\cdot \mid s_t) \longleftrightarrow o \sim \text{LLM}(\cdot \mid s_t)$, $a = \mathcal{P}(o)$, where $\mathcal{P}$ is a rule-based mapping that selects the most frequently occurring action in the LLM output $o$.

An Example of LLM output and rule-based mapping is given as:

> **For ALFWorld(Pick) from Qwen2.5 7B**
>
> **Input:** Task: Your task is to: put some alarmclock on desk. Current observation:You turn on the desklamp 1.. You are allowed to take the following actions: examine desk 1, examine keychain 3, go to bed 1, go to drawer 1, go to drawer 2, go to drawer 3, go to drawer 4, go to drawer 5, go to dresser 1, go to garbagecan 1, inventory, look, put keychain 3 in/on desk 1, use desklamp 1.
> Please select an action from the admissible actions. Please just output the selected action:
> **LLM Output to Action:** 1: put keychain 3 in/on desk → put keychain 3 in/on desk
> 2: examine keychain 3 → examine keychain 3
> 3: I will choose "put keychain 3 in/on desk" → put keychain 3 in/on desk
> 4: You just picked up the desklamp 1 → Randomly select an action, since no feasible action appears in the output
> 5: use desklamp 1 → use desklamp 1
> We map the LLM's output to an executable action using a simple rule-based method, extracting the executable actions directly from the LLM's output.

### 10.3 COMPUTATIONAL RESOURCES AND HYPERPARAMETERS

Our experiments are conducted on a single machine equipped with eight 48 GB A6000 GPUs, using PyTorch 2.1 with CUDA 12.4. Tables 2, 3, 4, and 5 report the main hyperparameters of our algorithms. For the likelihood, we approximate it as $p(\mathcal{O} = 1 \mid s, a) \propto \exp(Q(s,a)/\tau)$, where the hyperparameter $\tau$ is used in two contexts: (1) value-function-guided action posterior selection, and (2) reweighting the log-likelihood during LLM prior fine-tuning, as shown in Eq. 7. In practice, we tune these two parameters separately and denote them as $\tau_1$ and $\tau_2$, respectively. The horizon and memory capacity of environments are shown in Table 6.

Table 2: The hyperparameters on Overcooked(Tomato )

| Baselines | Learning Rate | Epochs | Batch Size | Update Frequency | $\tau_1$ | $\tau_2$ | K | $\gamma$ |
|---|---|---|---|---|---|---|---|---|
| Mem-EM$_{\text{w/ tune}}$ | 5e-4 | 3 | 16 | 10 | 0.1 | 0.5 | 5 | 0.8 |

Table 3: The hyperparameters on Overcooked(Salad)

| Baselines | Learning Rate | Epochs | Batch Size | Update Frequency | $\tau_1$ | $\tau_2$ | K | $\gamma$ |
|---|---|---|---|---|---|---|---|---|
| Mem-EM$_{\text{w/ tune}}$ | 5e-4 | 3 | 16 | 10 | 0.1 | 0.5 | 10 | 0.8 |

Table 4: The hyperparameters on ALFWorld(Pick)

| Baselines | Learning Rate | Epochs | Batch Size | Update Frequency | $\tau_1$ | $\tau_2$ | K | $\gamma$ |
|---|---|---|---|---|---|---|---|---|
| Mem-EM$_{\text{w/ tune}}$ | 5e-4 | 3 | 16 | 100 | 0.1 | 0.2 | 5 | 0.9 |

Table 5: The hyperparameters on ALFWorld(Examine)

| Baselines | Learning Rate | Epochs | Batch Size | Update Frequency | $\tau_1$ | $\tau_2$ | K | $\gamma$ |
|---|---|---|---|---|---|---|---|---|
| Mem-EM$_{\text{w/ tune}}$ | 5e-4 | 3 | 16 | 100 | 0.1 | 0.5 | 10 | 0.9 |

## 11 ADDITIONAL RESULTS

### 11.1 ADDITIONAL BASELINES

To further demonstrate the sample efficiency of our memory-driven self-improvement framework, we compare our method with the policy-based LLM-tuning approach PPO (Tan et al., 2024). In

|  | ALFWorld(Pick) | ALFWolrd(Examine) | Cook(Tomato) | Cook(Salad) |
|---|---|---|---|---|
| Horizon | 60 | 60 | 15 | 30 |
| Memory Capacity | 100 | 1000 | 15000 | 15000 |

Table 6: Maximum horizon and memory capacity of environments.

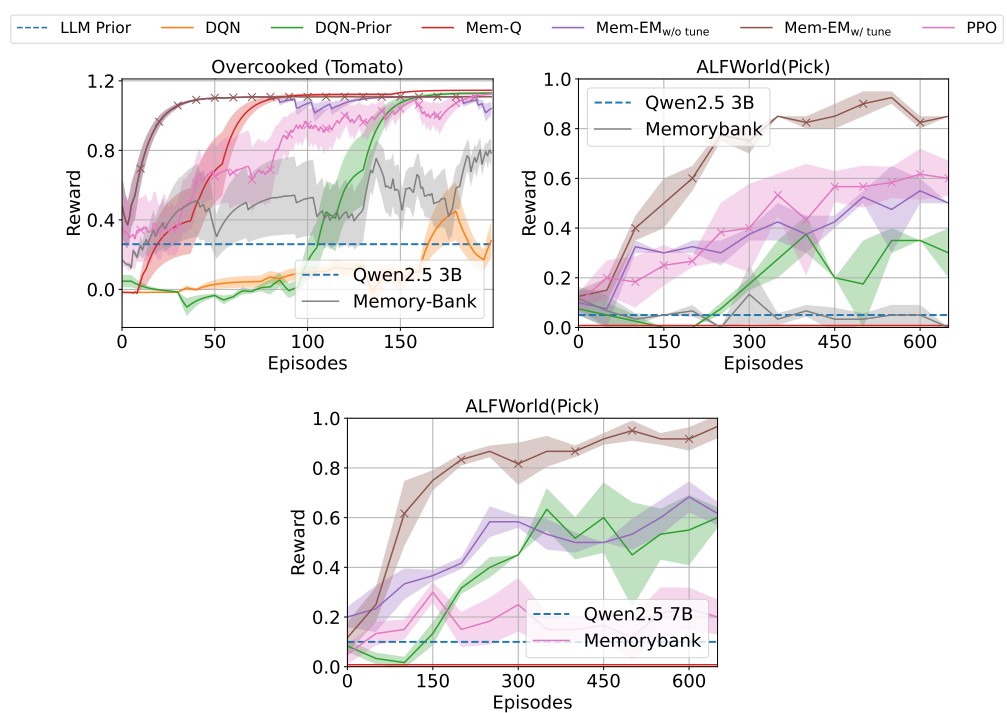

Figure 6: Additional Results of comparison with PPO. We plot the mean and standard error of the cumulative reward. The dashed line represents directly prompting the LLM prior to generating actions given state information, with the corresponding LLM version specified. The '×' markers for Mem-EM$_{w/\,tune}$ and PPO indicate the time steps when the LLM prior is fine-tuned. The baselines in each subfigure are run on the identical foundation LLM, which is explicitly noted in the legend.

this baseline, the LLM acts as the action policy and is fine-tuned following the PPO algorithm. As shown in Figure 6, although both methods involve fine-tuning LLMs, our Mem-EM$_{w/tune}$ significantly outperforms PPO thanks to the memory storage and value-guided action policy as described in Sec. 4.1.

We also compare with a memory-based LLM agent Memorybank (Zhong et al., 2024), for which previous state-action pairs are stored in memory, and the LLM makes decisions based on retrieved state-relevant memory. Although both our Mem-EM$_{w/tune}$ and MemoryBank involve memory for previous experiences, our Mem-EM$_{w/tune}$ outperforms MemoryBank on Overcooked(Tomato). The key difference lies in how memory is utilized: our method uses memory for value estimation and LLM prior refinement combined with a value-guided action policy, whereas MemoryBank simply uses retrieved memory as additional decision context for the LLM. On the more complex ALFWorld environment, we retrieved K=20 similar (s,a) pairs as input context to enhance the decision-making ability of the original LLMs (the same setting as our Mem-EM method). As shown in Figure 6, the Memorybank baseline, which uses retrieval memory as an enhanced prompt, performs significantly worse than our Mem-EM approach, which utilizes memory for value estimation and efficient search guidance. Furthermore, the Memorybank with Qwen2.5-3B performs similarly to the original base model, only showing better results when scaled up to Qwen2.5-7B. This clearly indicates that the effectiveness of the Memorybank method is highly limited by the foundation model size.

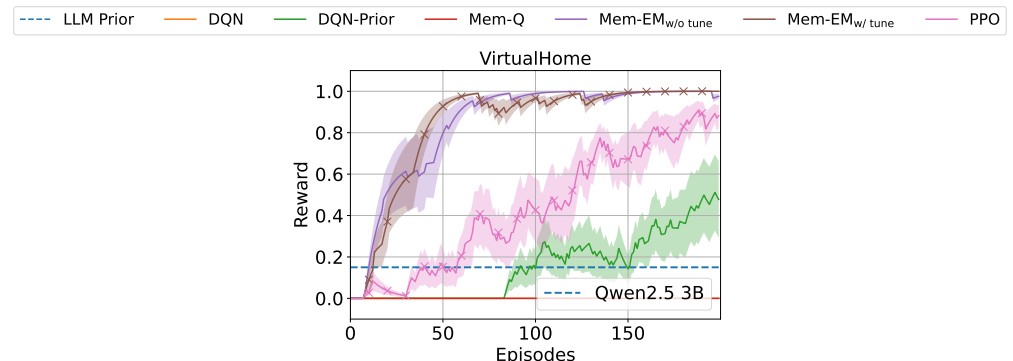

Figure 7: Additional Results on VirtualHome. We plot the mean and standard error of the cumulative reward. The dashed line represents directly prompting the LLM prior to generating actions given state information, with the corresponding LLM version specified. The '×' markers for Mem-EM$_{\text{w/ tune}}$ and PPO indicate the time steps when the LLM prior is fine-tuned.

|  | Without Finetune | Finetuned |
|---|---|---|
| Qwen2.5-3B | 0.05 | 0.85 |
| Qwen2.5-7B | 0.10 | 0.88 |
| LLaMA3-8B | 0.0 | 0.83 |

Table 7: Results of fine-tuning different LLMs on ALFWorld(Pick). All models are fine-tuned with the same hyperparameters: learning rate of 5e-4, LoRA rank of 16, and 5 epochs.

## 11.2 ADDITIONAL BENCHMARK

To further verify the effectiveness of our framework, we consider an additional text-based benchmark, VirtualHome (Puig et al., 2018), which is widely used to evaluate LLM-based decision-making (Tan et al., 2024; Wen et al., 2024). Following prior work, we consider the food preparation task, which requires the agent to complete the task within 4 rooms. The reward is sparse, i.e., the agent receives a reward of 1 only upon completing the target and 0 otherwise. The action space consists of up to 10 possible actions. As shown in Figure 7, due to the sparse reward signal, DQN and Mem-Q are unable to successfully explore. Methods that involve LLMs can incorporate LLM prior knowledge into the exploration process, thus performing better than the LLM-free DQN and Mem-Q baselines. In addition, our Mem-EM demonstrates superior sample efficiency and is more robust than PPO and DQN-Prior.

## 11.3 TESTING THE FRAMEWORK WITH DIFFERENT FOUNDATION LLMS

To validate the effectiveness of our method across different foundation LLMs, we fine-tune three models: Qwen2.5-3B, Qwen2.5-7B, and LLaMA3-8B. We use the memory table obtained by performing Mem-EM$_{\text{w/tune}}$ with Qwen2.5-7B on ALFWorld(Pick), collecting approximately 6,000 $(s, a, Q)$ tuples. The three LLMs are then fine-tuned using our proposed LLM prior refinement objective function (Eq. 7) with this memory table. As shown in Table 7, our LLM fine-tuning objective is adaptable to different LLMs, which demonstrates that the memory gathered using Qwen2.5-7B can be easily distilled into both the smaller model Qwen2.5-3B and a model with a different architecture, LLaMA3-8B.

## 11.4 COMPUTATION RESOURCE REQUIREMENTS

Table 8 shows computation costs for the baselines in ALFWorld(Pick) with Qwen2.5-3B. Our method achieves significant performance gains over PPO and DQN-Prior while maintaining similar training time and GPU usage requirements as the previous best-performing method, PPO.

| | PPO | DQN | DQN-Prior | Mem-EM$_{w/\,tune}$ | Mem-EM$_{w/o\,tune}$ |
|---|---|---|---|---|---|
| Performance | 0.62 | 0 | 0.38 | 0.93 | 0.55 |
| Training Time | 11.05h | / | 12.07h | 11.20h | 13.57h |
| GPU usage | 13661M | / | 8991M | 15823M | 8013M |
| Memory table size | / | / | / | 7100 | 7554 |
| Memory CPU usage | / | / | / | 57.42MB | 61.12MB |
| Inference Time(query 1000 times)/minutes | 3.40 | / | 3.59 | 5.28 | 8.16 |

Table 8: Computational Resource on ALFWorld(Pick) with Qwen2.5-3B.

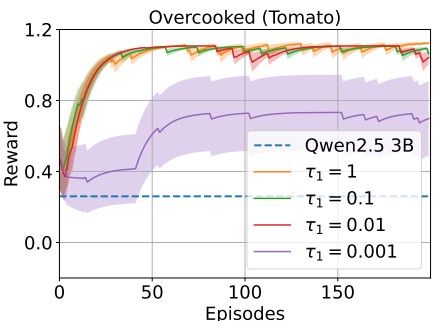

Figure 8: Ablation on the exploration hyperparameter $\tau_1$ of Mem-EM$_{w/o\,tune}$ on the Overcooked(Tomato) with Qwen2.5-3B.

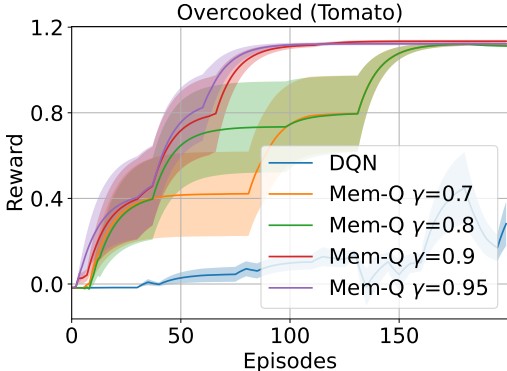

Figure 9: Ablation on the exploration hyperparameter discount factor $\gamma$ of our memory-based Q estimator Mem-Q on the Overcooked(Tomato).

## 11.5 ADDITIONAL RESULTS ON THE HYPERPARAMETER TUNING

**Ablation on exploration trade-off hyperparameter** We have conducted an experiment on the exploration trade-off hyperparameter $\tau_1$. Specifically, we tuned this hyperparameter of our Mem-EM$_{w/o\,tune}$ on Overcooked(Tomato) with Qwen2.5-3B. As shown in Figure 8, the hyperparameter $\tau_1$ is not as sensitive as we imagined, and the algorithm fails to learn only under the setting of $\tau_1 = 0.001$, which is extremely greedy and poor for exploration.

**Ablation on discount factor $\gamma$** We conduct an ablation study on the discount factor $\gamma$ using Overcooked(Salad) with dense rewards. As shown in Figure 9, our Mem-Q demonstrates robustness to different values of $\gamma$.

**Additional ablation on the memory table capacity** We conduct an ablation study on the memory table capacity of Mem-EM$_{w/tune}$ on Overcooked(Salad) with Qwen2.5-7B As shown in Figure 10, our Mem-EM$_{w/tune}$ demonstrates robustness to limited memory capacity. This robustness is attributed to the LLM's role in reducing the exploration space, allowing the model to converge ef-

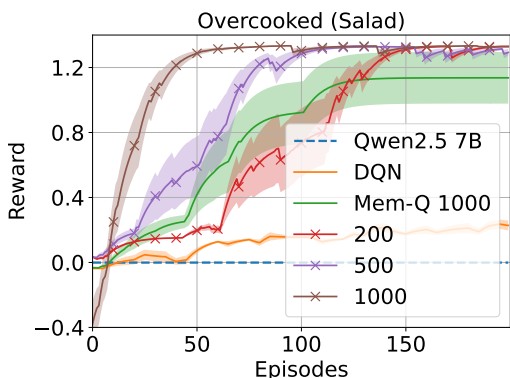

Figure 10: Ablation on the memory table capacity of Mem-EM$_{\text{w/tune}}$ on Overcooked(Salad) with Qwen2.5-7B.

fectively with only 200 memory items, despite a total of approximately 1,000 possible state-action pairs in the environment.

## LLM USAGE STATEMENT

In this work, we used large language models (LLMs) to assist with writing and polishing the manuscript. Specifically, LLMs were employed to improve the grammar, clarity, and overall readability of the text. All scientific content, ideas, and experimental results were generated and verified by the authors. The use of LLMs was limited strictly to language editing, and no LLMs were used to generate research content, data analysis, or experimental results.

