# OpenReview forum: "Memory-Driven Self-Improvement with Large Language Models"
_ICLR.cc/2026/Conference — ICLR 2026 Conference Withdrawn Submission_

### Official Review · Reviewer_po9h · 2025-10-27

**Soundness:** 2
**Presentation:** 3
**Contribution:** 1
**Rating:** 2
**Confidence:** 3

**Summary:**

The paper addresses the limiation in efficiently adapting LLMs to text-based decision-making tasks, and proposes a method to combine LLM priors with task-specific interactions.

In this method, the LLM prior generates action candidates and memory-driven value estimation guides action selection, establishing a self-imporvement loop between the fine-tuned LLM priors (through LoRA) and the updated value estimation (through Q-table).

Experiments on 2 tasks in Overcooked and 2 tasks in ALFWorld shows the effectiveness of proposed method compared to traditional RL method DQN (and its variants) and fixed LLM priors (and its variants).

**Strengths:**

1. Memory is a key aspect of LLM for improved performance especially in agent tasks. This paper proposes a memory mechanism that stores Q-values for visited state–action pair, which aims to select more precise actions and refine the LLM parameters.
2. Experiments show effective self-reinforcing effects between updated memory table and refined LLM priors compared to DQN and variants of Mem-EM.

**Weaknesses:**

1. **Insufficient baselines to compare.**
The compared baselines in the paper only include DQN variants and variants of the proposed method (Mem-EM).
Many other approaches (e.g., [1][2][3]) incorporating memory into LLM agents are not compared. Thus, current results cannot demonstrate the effectiveness of proposed Q-memory compared to other forms of memory for LLM agents.
2. **Insufficient benchmark experiments raises concern about the generalization of the proposed method across different domains and tasks.**
The paper only chooses 2 tasks from Overcooked and 2 tasks from ALFWorld. Why not consider all the task set in these two environments? Is there any special reason to choose these four tasks in experiments?
3. **Concerns about the practicality of the proposed method.**
Mem-EM needs to store memory table, which is impractical when facing open-ended environments with infinite state/action spaces.
Additionally, Mem-EM needs to compare Q-values of different action candidates for the same state, however, in many text-based environments (e.g, dialogue) we cannot find the same intermediate state within a trajectory.
4. **Issues in both memory and compuation cost.**
Due to the memory table storage and multiple action sampling per state, Mem-EM induces more memory and LLM rollout costs despite it can improve exploration efficiency.

References:

[1] Xu, Wujiang, et al. "A-mem: Agentic memory for llm agents." arXiv preprint arXiv:2502.12110 (2025).

[2] Packer, Charles, et al. "MemGPT: Towards LLMs as Operating Systems." (2023).

[3] Zhong, Wanjun, et al. "Memorybank: Enhancing large language models with long-term memory." Proceedings of the AAAI Conference on Artificial Intelligence. Vol. 38. No. 17. 2024.

**Questions:**

1. What is the difference between $LLM_{Mem-EM}$ and $LLM_{Mem-EM}+Q_{Mem-EM_{w/tune}}$ in Table 1?
Better to provide the mean results and standard error in Table 1 across different seeds.
2. Why do $LLM_{Mem-EM}+Q_{Mem-EM_{w/tune}}$ in Table 1 show better performance when K=5 compared to larger K?
3. It is challenging to estimate value when process rewards are not available according to Eq.2. What is the value of $\gamma$ used in Eq.2? Any ablation studies for this?
4. Is it worth to improve exploration effciency at the cost of more memory and computation costs? Please provide detailed memory and computation costs before and after using Mem-EM.

Typo: citation format in the paper

---

> ### Author Response · Authors · 2025-11-23
>
> We thank the reviewer for their time and effort in reviewing our paper. We are glad that you appreciate: 1) our focus on using memory to improve LLM performance in agent tasks, and 2) our experimental results demonstrating effective self-reinforcing effects between the updated memory table and refined LLM priors. We have provided detailed explanations and clarifications to resolve your concerns and made several updates that have improved the paper based on your recommendation. We respectfully hope you can reconsider your score to reflect our updates.
>
> > Q1: **Insufficient baselines to compare**
>
> Thank you for your valuable suggestion. We have compared our method with the policy-based LLM fine-tuning approach **PPO** and a memory-based LLM agent **Memorybank**[1] as you recommended.
>
> Practically, in PPO baseline, the LLM acts as the action policy and is fine-tuned following the PPO algorithm. In the MemoryBank baseline, previous state-action pairs are stored in memory, and the LLM makes decisions based on retrieved state-relevant memory.
>
> As shown in the table below (see Figure 6 in Appendix for more details), although both methods involve fine-tuning the same LLMs, our **$\text{Mem-EM}_\text{w/tune}$ outperforms PPO** thanks to the memory storage and value-guided action policy as described in section 4.1.
>
> Although both our  $\text{Mem-EM}_\text{w/tune}$ and MemoryBank involve memory for previous experiences, **our $\text{Mem-EM}_\text{w/tune}$ outperforms MemoryBank**. The key difference lies in how memory is utilized: our method uses memory for value estimation and LLM prior refinement combined with a value-guided action policy, whereas MemoryBank simply uses retrieved memory as additional decision context for the LLM.
>
> **Results on Overcooked(Tomato) with Qwen2.5-3B**
>
> | Baselines/Training Episodes     | 50       | 100      | 150      | 200      |
> | ------------------------------- | -------- | -------- | -------- | -------- |
> | $\text{Mem-EM}_\text{w/tune}$   | **1.01** | **1.11** | **1.11** | **1.11** |
> | $\text{Mem-EM}_\text{w/o tune}$ | 1.01     | 1.09     | 1.09     | 1.08     |
> | PPO                             | 0.48     | 0.81     | 0.98     | 1.08     |
> | Memory-bank                     | 0.40     | 0.51     | 0.52     | 0.65     |
> | DQN-Prior                       | -0.03    | 0.03     | 0.81     | 1.13     |
> | DQN                             | 0        | 0.07     | 0.09     | 0.29     |
>
> **Results on ALFWorld(Pick) with Qwen2.5-3B**
>
> | Baselines/Training Episodes     | 150      | 300      | 450      | 600      |
> | ------------------------------- | -------- | -------- | -------- | -------- |
> | $\text{Mem-EM}_\text{w/tune}$   | **0.40** | **0.78** | **0.82** | **0.93** |
> | $\text{Mem-EM}_\text{w/o tune}$ | 0.32     | 0.30     | 0.38     | 0.47     |
> | PPO                             | 0.18     | 0.38     | 0.43     | 0.58     |
> | DQN-Prior                       | 0.03     | 0.08     | 0.38     | 0.35     |
> | DQN                             | 0        | 0        | 0        | 0        |
>
> [1]Zhong, Wanjun, et al. "Memorybank: Enhancing large language models with long-term memory." Proceedings of the AAAI Conference on Artificial Intelligence. Vol. 38. No. 17. 2024.

---

> ### Author Response · Authors · 2025-11-23
>
> > Q2: **Insufficient benchmark experiments raises concern about the generalization of the proposed method across different domains and tasks**.
>
> We selected these four tasks following the evaluation protocols established in prior work [3] [4] to enable direct comparison with baseline methods. These tasks capture the key challenges present in both environments, including long-term planning, large exploration spaces, and sparse reward signals.
>
> To further address you concerns on insufficient benchmarks, we have added experiments on an **additional text-based benchmark, VirtualHome[1]**, which is widely used to evaluate LLM-based decision-making [2] [3]. Following prior work, we consider the food preparation task, which requires the agent to complete the task within 4 rooms. This task is challenging due to **partial observability and sparse rewards**: only objects in the current room are visible, the agent receives reward 1 only upon task completion (0 otherwise), and the action space contains up to 10 possible actions.
>
> As shown in the table below (see Figure 7 in Appendix for details), our Mem-EM significantly outperforms all other baselines. Due to the sparse reward signal and partial observability, **traditional RL algorithms, including DQN and Mem-Q, are unable to successfully explore**. **Methods that involve LLMs** can incorporate rich prior knowledge into the exploration process, thus performing better than the LLM-free DQN and Mem-Q baselines. In addition, our **$\text{Mem-EM}_\text{w/tune}$ demonstrates superior sample efficiency and is more robust** than PPO and DQN-Prior.
>
> **Results on VirtualHome with Qwen2.5-3B**
>
> | Baselines/Training Episodes     | 50       | 100      | 150      | 200      |
> | - | - | -| --| -|
> | $\text{Mem-EM}_\text{w/tune}$   | **0.61** | 0.95     | 0.97 | **1.00** |
> | $\text{Mem-EM}_\text{w/o tune}$ | 0.58     | **0.98** | **0.99** | 0.99     |
> | PPO   | 0.09     | 0.35     | 0.61     | 0.84     |
> | DQN-Prior  | 0.00| 0.07     | 0.21     | 0.42     |
> | DQN  | 0  | 0   | 0   | 0 |
> | Mem-Q  | 0 | 0  | 0| 0        |
>
>
>
> [1]Puig, Xavier, et al. "Virtualhome: Simulating household activities via programs." *Proceedings of the IEEE conference on computer vision and pattern recognition*. 2018.
>
> [2] Wen, Muning, et al. "Reinforcing llm agents via policy optimization with action decomposition." *Advances in Neural Information Processing Systems* 37 (2024): 103774-103805.
>
> [3] Tan, Weihao, et al. "True Knowledge Comes from Practice: Aligning Large Language Models with Embodied Environments via Reinforcement Learning." *The Twelfth International Conference on Learning Representations*.
>
> [4] Yan, Xue, et al. "Efficient Reinforcement Learning with Large Language Model Priors." *The Thirteenth International Conference on Learning Representations*.
>
> > Q3:**Concerns about the practicality of the proposed method**.
>
>
> Thank you for raising these important concerns regarding the practicality of Mem-EM, particularly concerning scalability to open-ended environments and same-state comparisons.
>
> **1. Regarding open-ended environments with vast or infinite state/action spaces**
>
> We argue that the design of the Mem-EM framework is highly adaptable to open-ended environment settings, supported by the following evidence:
>
> - **Kernel-based value estimation:** Our kernel-based Q-value estimation relies on semantic similarity retrieval rather than exact matching. For unseen states or actions, the method retrieves semantically similar state-action pairs stored in memory to perform Q-value estimation as shown in Eq. 3, enabling effective generalization to infinite state/action spaces.
> - **Empirical evidence:** ALFWorld features a huge state space and state-varying large action spaces. Our superior performance on this benchmark empirically demonstrates our method's scalability to vast state/action spaces.
> - **Memory efficiency:** Our ablation studies on memory table capacity (Figure 4(c)) show that our method successfully converges even with limited memory capacity and a simple LRU (Least Recently Used) replacement mechanism. This confirms that the method does not require storing every single state-action pair but efficiently leverages a compact subset of frequently-accessed experiences.
>
> **2.Regarding the comparison of Q-values for the same state:**
>
> We would like to clarify that this concern do not affect the practicality of our method, because:
>
> - **Global Memory Store:** the memory table accumulates experiences across the entire training stage, not just single trajectories, enabling cross-trajectory matching.
>
> - **Kernel-based generalization:** As stated above, our method relies on kernel-based value estimation and generalization across the semantic space rather than strict exact state matching.

---

> ### Author Response · Authors · 2025-11-23
>
> > Q4:**Issues in both memory and compuation cost.**
>
> Thank you for your valuable suggestions. We have added detailed computation costs of the baselines on ALFWorld(Pick) with Qwen2.5-3B, as shown in the table below. Our method achieves significant performance gains over PPO and DQN-Prior while maintaining similar training time and GPU usage requirements as the previous best-performing method, PPO.
>
> |                                          | PPO     | DQN-Prior | $\text{Mem-EM}_{\text{w/tune}}$ | $\text{Mem-EM}_{\text{w/o tune}}$ |
> | ---------------------------------------- | ------- | --------- | ------------------------------- | --------------------------------- |
> | Performance                              | 0.62    | 0.38      | 0.93                            | 0.55                              |
> | Training Time                            | 11.05h  | 12.07h    | 11.20h                          | 13.57h                            |
> | Training GPU Usage                       | 13.34GB | 8.78GB    | 15.45GB                         | 7.8GB                             |
> | Memory CPU Usage                         | /       | /         | 57.42MB                         | 61.12MB                           |
> | Memory Table Size                        | /       | /         | 7100                            | 7554                              |
> | Inference time(query 1000 times)/minutes | 3.40    | 3.59      | 5.28                            | 8.16                              |
>
> > Q5: What is the difference between $\text{LLM}/{\text{Mem-EM}\text{w/tune}}$ and $\text{LLM}/{\text{Mem-EM}\text{w/tune}}+Q_{\text{Mem-EM}\text{w/tune}}$  in Table 1? Better to provide the mean results and standard error in Table 1 across different seeds
>
> The $\text{Mem-EM}\text{w/tune}$ generates a refined LLM and a memory table of $(s,a,Q)$ tuples. The $\text{LLM}/{\text{Mem-EM}\text{w/tune}}$  uses the refined LLM to directly generate a single action for execution. The $\text{LLM}/{\text{Mem-EM}\text{w/tune}}+Q_{\text{Mem-EM}\text{w/tune}}$  uses the refined LLM to generate $K$ action candidates and then selects one action based on value estimation following the memory table. We will add the mean and standard error to Table 1 as per your suggestion.
>
> > Q6: Why do $\text{LLM}/{\text{Mem-EM}\text{w/tune}}+Q_{\text{Mem-EM}\text{w/tune}}$ in Table 1 show better performance when K=5 compared to larger K?
>
> Thank you for your question. The better performance when K=5 (following the training setting) compared to larger K may **stem from the introduction of low-quality, noisy samples from the tail distribution of the refined LLM prior**. Specifically:
>
> 1. The refined LLM has distilled domain-specific decision-making knowledge and is capable of generating high-quality actions when  $K=5$. **Case Study Example:** In the state "You are carrying: a tissuebox 1..." with the task "put some tissuebox on coffeetable," the $K=5$ action candidates (`['put tissuebox 1 in/on coffeetable 1', 'inventory', 'put tissuebox 1 in/on coffeetable 1', 'examine coffeetable 1', 'examine coffeetable 1']`) successfully include the optimal action (`put tissuebox 1 in/on coffeetable 1`).
> 2. However, when $K$ is increased, less relevant or noisy action candidates are more likely to be sampled from the refined LLM prior's low-probability tail. This inclusion of noisy actions, combined with the inherent generalization limitations of the value estimation model, degrades the final policy performance.
>
> **Overall, these results indicates that K=5 balances sufficient exploration coverage with high-quality action candidates.**
>
>
>
> > Q7: It is challenging to estimate value when process rewards are not available according to Eq.2. What is the value of $\gamma$ used in Eq.2? Any ablation studies for this?
>
> Thank you for your question. In fact, the benchmarks ALFWorld and VirtualHome (added in Q2) do not provide process rewards, where agents receive reward 1 only upon task completion and 0 otherwise. Our experimental results demonstrate that our method successfully handles such sparse reward settings. As per your suggestion, we have listed values of the discount factor $\gamma$ in Section 10.3 of the Appendix, and Figure 9 shows an ablation study on $\gamma$ for Overcooked(Salad), demonstrating that our memory-based Q-estimation is robust across different $\gamma$ values.

---

> > ### Comment · Reviewer_po9h · 2025-11-27
> >
> > Thanks for the response. However, most of the weaknesses have not been addressed in my view.
> >
> > 1. Regarding the baslines:
> > - The added baseline MemoryBank is only compared on one task (Overcooked Tomato), which cannot demonstrate that it generally underperforms the proposed method.
> > - There are many recent approaches that incorporate memory into LLM agents as suggested above and Memorybank is just one of the simpler methods among them. So it would be better to compare with more memory-based baselines.
> > - Performance of PPO and MemoryBank provided in A1 consistently improve as training episodes increase, hence more training episodes might flip the order of performance between the added baseline and Mem-$EM_{w/ tune}$.
> > - Results of PPO on ALFWorld Pick in A1 does not match that in A4.
> > 2. Regarding the computation cost:
> > -  Mem-$EM_{w/ tune}$ consumes even silghtly more training time and GPU usage than PPO and more inference time, which raises concern when applying to larger language models in practice.
> > 3. Presentation:
> > - Ablation studies in Figure 4c and Figure 9 examines Mem-Q, but I think the main method the paper proposes is Mem-$EM_{w/ tune}$. So it is better to conduct all the ablation studies on Mem-$EM_{w/ tune}$.
> > - It would be better to clearly state which experiments use Qwen2.5-3B/Qwen2.5-7B (maybe in captions of each table or figure).
> > - The caption of Figure 9 is the same with that of Figure 8.
> > 4. One more question: In A6, the authors claim the inherent generalization limitations of the value estimation model. Is this one of the bottlenecks of Mem-$EM_{w/ tune}$ and how to mitigate this?
> >
> > Given the concerns above, I am keeping my score.

---

> > > ### Author Response · Authors · 2025-12-04
> > >
> > > ### Regarding baselines:
> > >
> > > > Q1: The added baseline MemoryBank is only compared on one task (Overcooked Tomato), which cannot demonstrate that it generally underperforms the proposed method.
> > >
> > > We have now incorporated the Memorybank method as a baseline on the complex ALFWorld task. Our results demonstrate the superior performance of our proposed method, Mem-EM, over the Memorybank approach (see Appendix 11.1 for details).
> > >
> > > Furthermore, we observed that the Memorybank implementation using Qwen2.5-3B dosen't show clearly performance gains, only showing improvement when scaled up to Qwen2.5-7B. This clearly indicates that the effectiveness of the Memorybank method is highly dependent on and limited by the size of the underlying foundation model.
> > >
> > >  **Results on ALFWorld(Pick) with Qwen2.5-3B**
> > >
> > > | Baselines/Training Episodes     | 150      | 300      | 450      | 600      |
> > > | ------------------------------- | -------- | -------- | -------- | -------- |
> > > | $\text{Mem-EM}_\text{w/tune}$   | **0.40** | **0.78** | **0.82** | **0.93** |
> > > | $\text{Mem-EM}_\text{w/o tune}$ | 0.32     | 0.30     | 0.38     | 0.47     |
> > > | PPO                             | 0.18     | 0.38     | 0.43     | 0.58     |
> > > | DQN-Prior                       | 0.03     | 0.08     | 0.38     | 0.35     |
> > > | DQN                             | 0        | 0        | 0        | 0        |
> > > | Memory-bank(Qwen2.5-3B)         | 0.03     | 0.00     | 0.07     | 0.05     |
> > > | Memory-bank(Qwen2.5-7B)         | 0.15     | 0.18     | 0.15     | 0.22     |
> > >
> > >
> > >
> > > > Q2: There are many recent approaches that incorporate memory into LLM agents as suggested above and Memorybank is just one of the simpler methods among them. So it would be better to compare with more memory-based baselines.
> > >
> > > We chose Memorybank as a **fair and representative comparison point** because it shares a classic memory usage method and is adaptive to decision-making tasks. Specifically, Memorybank, along with the other memory-based prompt engineering works you recommended, **share the core component of historical retrieval for enhanced generation**. However, the other two methods often require complex, task-specific designs. For instance, *Agentic Memory*involves dynamic memory evolution—where "the system determines whether to update its context, keywords, and tags." Designing appropriate keywords and tags for general decision-making tasks is non-trivial, and the method is tailored for long-context dialogue and QA tasks. Conversely, Memorybank's memory set, which uses historical state-action $(s, a)$pairs, is naturally adaptive for sequential decision-making tasks and shares the same memory storage method as ours. As shown in Q1, our Mem-EM approach shows superior performance than Memorybank due to the effective design of memory usage.
> > >
> > >
> > >
> > > > Q3:  Performance of PPO and MemoryBank provided in A1 consistently improve as training episodes increase, hence more training episodes might flip the order of performance between the added baseline and Mem-$EM_{w/ tune}$.
> > >
> > > With a limited number of training episodes, our method quickly converges to successful task completion on Overcooked and achieves a superior success rate (around 0.9) on ALFWorld tasks. This clearly demonstrates the **superior sample efficiency** of our Mem-EM approach. Sample efficiency is paramount in the decision-making domain, as the requirement for extensive training episodes is impractical and often **should be avoided** for complex, real-world applications.
> > >
> > > > Q4: Results of PPO on ALFWorld Pick in A1 does not match that in A4.
> > >
> > > For A1, we report the performance observed **during the training process**, while for A4, we report the **test-time performance** of the saved model. The slight performance difference observed is likely due to **randomness**.

---

> ### Author Response · Authors · 2025-12-04
>
> ### Regarding computation cost
> > Mem-EM consumes even silghtly more training time and GPU usage than PPO and more inference time, which raises concern when applying to larger language models in practice.
>
>
>
> To further highlight the computational efficiency of our method, we would like to make two claims. First, the direct deployment of the refined LLM prior, following $\text{Mem-EM}\text{w/tune}$, shows superior performance and the same inference overhead as PPO. Although the value-guided search increases the inference time of our approach by letting the refined LLM generate $\mathcal{A}$ action candidates and selecting one according to the Q-values, we can also directly apply the refined LLM by letting it generate one action for each state during inference, noted as LLM-$\text{Mem-EM}_\text{w/tune}$  (as explained in above Q5 and in lines 413 to 416).
>
>
>
> |                                          | PPO     | DQN-Prior | $\text{Mem-EM}_{\text{w/tune}}$ | LLM-$\text{Mem-EM}_{\text{w/tune}}$ | $\text{Mem-EM}_{\text{w/o tune}}$ |
> | ----------- | ------- | --------- | ----------- | ----------------------------------- | --------------------------------- |
> | Performance                              | 0.62    | 0.38      | 0.93                            | 0.83                                | 0.55                              |
> | Training Time                            | 11.05h  | 12.07h    | 11.20h                          | 11.20h                              | 13.57h                            |
> | Training GPU Usage                       | 13.34GB | 8.78GB    | 15.45GB                         | 15.45GB                             | 7.8GB                             |
> | Memory CPU Usage                         | /       | /         | 57.42MB                         | 57.42MB                             | 61.12MB                           |
> | Memory Table Size                        | /       | /         | 7100                            | 7100                                | 7554                              |
> | Inference time(query 1000 times)/minutes | 3.40    | 3.59      | 5.28                            | 3.45                                | 8.16                              |
>
> In addition, when adapting to large LLMs, our memory table is easily **distilled into smaller LLMs for more efficient deployment**. We added an experiment on **distilling memory collected from Qwen2.5-7B into Qwen2.5-3B**, following our memory-driven LLM refinement. In practice, we use the memory table obtained by performing $\text{Mem-EM}_{\text{w/tune}}$ with Qwen2.5-7B on ALFWorld(Pick), collecting approximately 6,000 $(s,a,Q)$ memory tuples. The LLMs are then fine-tuned using our proposed LLM prior refinement objective function (see Eq. 7) with this memory table. As shown in the table below, the **two LLMs achieve significant performance gains with our LLM fine-tuning objectives**, demonstrating that the decision-making knowledge reflected in the memory table can be successfully distilled into both the smaller model Qwen2.5-3B for more effective deployment.
>
> ### Regarding Presentation
>
> Q: Ablation studies in Figure 4c and Figure 9 examines Mem-Q, but I think the main method the paper proposes is ${Mem-EM}{w/tune}$. So it is better to conduct all the ablation studies on ${Mem-EM}_{w/tune}$.
>
> For Fig 4c on memory table capacity, we conduct an ablation study on the memory table capacity of $\text{Mem-EM}\text{w/tune}$using Qwen2.5-7B on the Overcooked(Salad) environment. As shown in Figure 10, our $\text{Mem-EM}_\text{w/tune}$ demonstrates robust convergence to limited memory capacity, which attributes to the LLMs' role in effectively narrowing down the exploration space.
>
> Figure 9 presents an ablation study on the reward discount factor, $\gamma$. Since $\gamma$ is exclusively involved in the memory-driven Q-value estimation component of our method, we believe this ablation, on $\text{Mem-Q}$ exploring the whole action space without LLMs, effectively and clearly illustrates the parameter's influence on the Q-value approximation. We have highlighted the specific LLM used in each subfigure, as you suggested.
>
> ### Regrading" One more question: In A6, the authors claim the inherent generalization limitations of the value estimation model. Is this one of the bottlenecks of Mem-EM and how to mitigate this?"
>
> Value estimation error, particularly when **generalizing to unseen tasks or states**, is **unavoidable for all existing methods**, including DQN, MCTS, our memory-based estimation, and LLM-as-judge approaches. This work focuses on **improving Q-value estimation** by using **vectorized memory retrieval** to enhance Q-value prediction for unseen state-action pairs. As demonstrated in Figure 2, Figure 3, and Table 1, our method with retrieval-based value estimation shows **superior performance** compared to DQN-based methods, indicating the advantages of our Q-value estimation in addressing these generalization challenges.

---

### Official Review · Reviewer_AnoS · 2025-10-27

**Soundness:** 3
**Presentation:** 3
**Contribution:** 2
**Rating:** 6
**Confidence:** 4

**Summary:**

This paper proposes a memory-driven self-improvement framework for text-based sequential decision-making that combines LLM general knowledge with a compact memory of domain-specific experiences. The framework consists of two components: (1) memory-driven value estimation using kernel-based Q-value approximation with LLM embeddings, and (2) memory-driven LLM prior refinement via an Expectation-Maximization formulation. The authors claim their approach achieves improvements on both in-distribution and out-of-distribution tasks.

**Strengths:**

- The paper presents an interesting combination of memory-based Q-learning with LLM priors in the EM framework.
- The paper is well written and easy to follow.

**Weaknesses:**

Please see my detailed questions and concerns below.

**Questions:**

- Given that you're using Qwen2.5 as your LLM prior, why not use the same LLM's encoder for state-action embeddings? Wouldn't this create better alignment between the value estimation and policy components?
- How do you handle the cold-start problem? When memory is empty or sparse early in training, how reliable are your Q-value estimates? This could significantly impact early exploration.
- How do you ensure memory diversity? The LRU replacement strategy might keep frequently visited states but miss important rare states.
- When K increases beyond training settings, performance often drops in unseen tasks (Table 1). Doesn't this suggest overfitting to the training distribution?
- Both test environments seem fully observable. How would this extend to POMDPs where memory of past states might be critical?

---

> ### Author Response · Authors · 2025-11-23
>
> We thank the reviewer for their time and effort in reviewing our paper, we are glad that you appreciate: 1) the interesting EM framework design, and 2) the paper being well-written and easy to follow. We have provided detailed explanations and clarifications to resolve your concerns and made several updates that have improved the paper based on your recommendation. We respectfully hope you can reconsider your score to reflect our updates.
>
> > Q1 Given that you're using Qwen2.5 as your LLM prior, why not use the same LLM's encoder for state-action embeddings? Wouldn't this create better alignment between the value estimation and policy components?
>
> Thank you for your question. In our framework, the LLM action prior is used to **generate high-quality textual action candidates**; thus, we require it to be sufficiently powerful. The state-action embedding model's function is to **generate semantically meaningful embeddings for vectorized retrieval-based value estimation**. Results in Figure 2 show that the smaller BERT-base model is also capable of solving the text-based Overcooked environment, indicating it has sufficient semantic embedding power to support our Q-value estimation. Specifically, the computation time for our Mem-Q with BERT-base and Qwen2.5-3B is shown in the table below. Therefore, we select the BERT-based model as the state-action embedding model, which provides a good **trade-off between effective Q-value estimation and computational efficiency**.
>
> |                   | Training Time |
> | ----------------- | ------------- |
> | Mem-Q(Bert-base)  | 3.2 min       |
> | Mem-Q(Qwen2.5-3B) | 24.4 min      |
>
>
>
> > Q2: How do you handle the cold-start problem? When memory is empty or sparse early in training, how reliable are your Q-value estimates? This could significantly impact early exploration.
>
> Thank you for your question. Indeed, experimental results demonstrate that our method can achieve cold-start without specialized intervention. In our implementation, the memory table starts empty and gradually expands during the exploration process. Additionally, our method retrieves semantically similar (s,a) pairs to augment value estimation. As shown in Figure 2, incorporating more similar (s,a) embeddings in the kernel-based Q estimation leads to faster convergence. This is a potential reason why our method handles cold-start scenarios effectively
>
> > Q3: How do you ensure memory diversity? The LRU replacement strategy might keep frequently visited states but miss important rare states.
>
> Thank you for this important question. We argue that frequently visited states are often important as they correspond to critical decision points in the task. As demonstrated in Figure 4(c), our method performs well with a simple LRU replacement strategy even under small memory capacity. This is because LRU effectively retains task-relevant frequent states, while our semantic similarity-based retrieval potentially enables generalization to rare but similar states, thereby alleviating the concern about missing important rare experiences.
>
> > Q4: When K increases beyond training settings, performance often drops in unseen tasks (Table 1). Doesn't this suggest overfitting to the training distribution?
>
> Thank you for your question. As shown in Table 1, **the refined LLM significantly improves performance on unseen tasks compared to the original LLM**. Therefore, we argue that the **refined LLM prior does not overfit** to the training distribution.
>
> The observed performance drop when $K$ is increased beyond the training setting may **stem from the introduction of low-quality, noisy samples from the tail distribution of the refined LLM prior**. Specifically:
>
> 1. The refined LLM has distilled domain-specific decision-making knowledge and is **capable of generating high-quality actions when $K$ is small** (e.g., $K=5$). **Case Study Example:** In the state "You are carrying: a tissuebox 1..." with the task "put some tissuebox on coffeetable," the $K=5$ action candidates (`['put tissuebox 1 in/on coffeetable 1', 'inventory', 'put tissuebox 1 in/on coffeetable 1', 'examine coffeetable 1', 'examine coffeetable 1']`) successfully include the optimal action (`put tissuebox 1 in/on coffeetable 1`).
> 2. However, **when $K$ is increased, less relevant or noisy action candidates** are more likely to be sampled from the refined LLM prior's low-probability tail. This inclusion of noisy actions, combined with the inherent generalization limitations of the value estimation model, degrades the final policy performance.
>
> Overall, these results indicates that K=5 balances sufficient exploration coverage with high-quality action candidates.

---

> ### Author Response · Authors · 2025-11-23
>
> > Q5: Both test environments seem fully observable. How would this extend to POMDPs where memory of past states might be critical?
>
> Thank you for this valuable point. To further verify the effectiveness of our method under the POMDP setting, we have added experiments on an **additional POMDP benchmark, VirtualHome** [1] [2] [3]. Specifically, we consider the food preparation task, which requires the agent to complete the task within 4 rooms. This task is partially observable, as only the objects in the agent's current room are visible. The reward is sparse, and the action space consists of up to 10 possible actions at each step.
>
> As shown in the table below (see Figure 7 in Appendix for details), our **Mem-EM is capable of solving this partially observable task because the rich prior knowledge in LLMs** enables the selection of reasonable action candidates, thereby facilitating the handling of partial observability.
>
> **Case Study Example:**
>
> - **State:** "You are in the kitchen. The microwave is closed and opened. You are not grabbing anything."
> - **Admissible actions:** `['walk to the living room', 'walk to the bathroom', 'walk to the bedroom', 'walk to the pancake', 'walk to the microwave']`
> - **LLM-generated action candidates:** `['walk to the pancake', 'walk to the microwave']`
>
> LLM-generated **actions are task-relevant**, avoiding irrelevant actions such as walking to other rooms. This demonstrates that the LLM prior effectively filters the action space based on task context, thereby helping to address partial observability.
>
> **Results on VirtualHome with Qwen2.5-3B:**
>
> | Baselines/Training Episodes     | 50       | 100      | 150      | 200      |
> | ------------------------------- | -------- | -------- | -------- | -------- |
> | $\text{Mem-EM}_\text{w/tune}$   | **0.61** | 0.95     | 0.97     | **1.00** |
> | $\text{Mem-EM}_\text{w/o tune}$ | 0.58     | **0.98** | **0.99** | 0.99     |
> | PPO                             | 0.09     | 0.35     | 0.61     | 0.84     |
> | DQN-Prior                       | 0.00     | 0.07     | 0.21     | 0.42     |
> | DQN                             | 0        | 0        | 0        | 0        |
> | Mem-Q                           | 0        | 0        | 0        | 0        |
>
> [1]Puig, Xavier, et al. "Virtualhome: Simulating household activities via programs." *Proceedings of the IEEE conference on computer vision and pattern recognition*. 2018.
>
> [2] Wen, Muning, et al. "Reinforcing llm agents via policy optimization with action decomposition." *Advances in Neural Information Processing Systems* 37 (2024): 103774-103805.
>
> [3] Tan, Weihao, et al. "True Knowledge Comes from Practice: Aligning Large Language Models with Embodied Environments via Reinforcement Learning." *The Twelfth International Conference on Learning Representations*.
>
> ####

---

### Official Review · Reviewer_PotS · 2025-10-30

**Soundness:** 3
**Presentation:** 3
**Contribution:** 3
**Rating:** 6
**Confidence:** 3

**Summary:**

The paper proposes a "Memory-Driven Self-Improvement" framework to adapt Large Language Models (LLMs) for specific sequential decision-making (SDM) tasks, addressing the insufficiency of their general prior knowledge. The method combines the LLM's prior with a compact memory that stores domain-specific experiences, specifically past state-action pairs and their estimated Q-values. This framework establishes a mutually reinforcing loop: a memory-driven, non-parametric value estimator retrieves semantically similar experiences to guide efficient exploration, while an Expectation-Maximization (EM) algorithm periodically refines the LLM prior by learning from the high-quality interactions stored in memory. This refined prior then generates higher-reward trajectories, which in turn enriches the memory. Experiments in environments like ALFWorld and Overcooked demonstrate that this self-improvement cycle significantly outperforms traditional reinforcement learning and LLM-based baselines in both sample efficiency and generalization to unseen tasks.

**Strengths:**

*  **Novel Synergistic Framework:** The paper introduces a novel and principled framework that synergistically combines LLM priors with a memory-augmented Q-learning. This memory-driven approach, which stores and retrieves domain-specific `(s, a, Q)` experiences, allows the agent to effectively integrate the LLM's generalist knowledge with high-value, task-specific trajectories, leading to significant performance gains on complex benchmarks like ALFWorld.

*  **Superior Sample Efficiency:** A key advantage of this framework is its remarkable sample efficiency compared to traditional methods like reinforcement learning fine-tuning (RLFT).

*  **Thorough Empirical Ablation:** The paper's thorough empirical ablation studies demonstrate its robustness and generalization.

**Weaknesses:**

* The framework's performance appears highly sensitive to parameter tuning. Its multi-component design introduces numerous hyperparameters, suggesting that performance could vary significantly with different settings, potentially impacting robustness.

* The algorithm's effectiveness is critically dependent on the external embedding model. The core memory-driven value estimation is retrieval-based, and the embedding model's performance will significantly impact the algorithm's overall performance.

**Questions:**

* How does the time-efficiency of the framework at test time compare to baselines like DQN and DQN-Prior?

* The hyperparameter $\tau_1$ governs the stochasticity of the posterior action selection and thus directly manages the exploration-exploitation trade-off of the policy. Could the authors provide results demonstrating the algorithm's performance with different values of $\tau_1$?

---

> ### Author Response · Authors · 2025-11-19
>
> We thank the reviewer for their time and effort in reviewing our paper, we are glad that you appreciate 1)**Novel Synergistic Framework** , 2) **Superior Sample Efficiency** and 3) **Thorough Empirical Ablation** of our work. We have provided detailed explanations and clarifications to resolve your concerns and made several updates that have improved the paper based on your recommendation. We respectfully hope you can reconsider your score to reflect our updates.
>
> > Q1: Concern about “The framework's performance appears highly sensitive to parameter tuning”, especially about the hyperparameter $\tau_1$ that manages the exploration-exploitation trade-off of the policy.
>
> Thank you for your question. In the experiments section, we conducted ablation studies on the number of action candidates, the LLM update interval, and the memory table volume. These results demonstrate that our algorithm is robust to these hyperparameters.
> In addition, to further address your concerns, we have added an experiment on the **exploration trade-off hyperparameter $\tau_1$**. Specifically, we tuned this hyperparameter of our $\text{Mem-EM}_\text{w/o tune}$ on Overcooked(Tomato) with Qwen2.5-3B. As shown in the table below (see Fig. 8 in the appendix for details), the hyperparameter $\tau_1$ is not as sensitive as we imagined, and the algorithm **fails to learn only under the setting of $\tau_1=0.001$**, which is extremely greedy and poor for exploration.
>
> |                | Performance |
> | -------------- | ----------- |
> | $\tau_1$=1     | 1.10        |
> | $\tau_1$=0.1   | 1.09        |
> | $\tau_1$=0.01  | 1.08        |
> | $\tau_1$=0.001 | 0.70        |
>
> > Q2 The algorithm's effectiveness is critically dependent on the external embedding model, the embedding model's performance will significantly impact the algorithm's overall performance.
>
>
>
> As you noted, the results in Figure 2 align with your statement: generally, **larger LLMs used as embedding models lead to better performance**. However, for practical resource consideration, we leverage the **BERT-base model** (approximately 110M parameters) as the embedding model for our main results, which balances efficient computation and reliable semantic embedding. Our experimental results show that this smaller BERT-base model is also capable of solving the text-based Overcooked environment and demonstrates superior performance in complex ALFWorld task, indicating it has **sufficient semantic embedding power** to support our retrieval-based Q-value estimation.
>
> > Q3 About time-efficiency of the framework at test time
>
> Thank you for your valuable suggestion. We have added detailed computation costs during training and inference time of the baselines on ALFWorld(Pick) with Qwen2.5-3B, as shown in the table below. Our method achieves significant performance gains over PPO and DQN-Prior while maintaining tolerable time-efficiency.
>
> |                                          | PPO     | DQN-Prior | $\text{Mem-EM}_{\text{w/tune}}$ | $\text{Mem-EM}_{\text{w/o tune}}$ |
> | ---------------------------------------- | ------- | --------- | ------------------------------- | --------------------------------- |
> | Performance                              | 0.62    | 0.38      | **0.93**                        | 0.55                              |
> | Training Time                            | 11.05h  | 12.07h    | 11.20h                          | 13.57h                            |
> | Training GPU Usage                       | 13.34GB | 8.78GB    | 15.45GB                         | 7.8GB                             |
> | Memory CPU Usage                         | /       | /         | 57.42MB                         | 61.12MB                           |
> | Memory Table Size                        | /       | /         | 7100                            | 7554                              |
> | **Inference time(query 1000 times)/minutes** | 3.40    | 3.59      | 5.28                            | 8.16                              |
>
> ####

---

### Official Review · Reviewer_dA4m · 2025-11-01

**Soundness:** 2
**Presentation:** 2
**Contribution:** 2
**Rating:** 2
**Confidence:** 4

**Summary:**

The paper proposes a memory-driven self-improvement framework for text-based sequential decision making. It has two coupled pieces: (i) Mem-Q, a non-parametric, retrieval-based Q-estimator that stores past  (s,a,Q) in a memory table and estimates Q(s,a) via k-NN with an inverse-distance kernel over LLMembeddings; and (ii) Mem-EM, an EM-style procedure that improve the LLM policy. Experiments on Overcooked (text) and ALFWorld report faster learning than DQN/DQN-Prior and gains for Mem-EM with LoRA fine-tuning.

**Strengths:**

1. The paper is easy to follow: it combines non-parametric value estimation with control-as-inference type policy learning.

2. Studies over candidate count, memory size and embedding backbones help illuminate when the system works best

**Weaknesses:**

1. The Mem-Q piece is simply Episodic Control with better embeddings; Mem-EM resembles MPO/EM-style policy updates with a learned prior. The paper cites these lines but does not sharply distinguish what is algorithmically new (beyond swapping in LLM embeddings and an LLM prior). In other words, it seems to me that the paper mainly combines these two ideas in the LLM domain. New insights when LLM coming into the picture is lacking.

2. Off-policy/Q-type-learning convergence is typically unstable. The paper even relies on importance sampling. Furthermore, the importance sampling weight is a heuristic approximation. Finally, does the memory table has sufficient coverage/exploratory enough? This makes people questions the practicality of the algorithms. I believe when we talk about policy optimization for LLMs, we have to be careful and understand what is new/different compared with standard DRL setting.

3. Baselines studies are also limited. For example, no comparisons with search-based planners guided by value, no comparisons with more advanced memory-based RL algorithms.

**Questions:**

See above

---

> ### Author Response · Authors · 2025-11-19
>
> We thank the reviewer for their time and effort in reviewing our paper, we are glad that you appreciate that 1)our paper is easy to follow , 2) effective ablation studies. We have provided detailed explanations and clarifications to resolve your concerns and made several updates that have improved the paper based on your recommendation. We respectfully hope you can reconsider your score to reflect our updates.
>
> ### **Weakness 1: Lack of Algorithmic Novelty and Missing Insights with LLMs**
>
> We thank the reviewers for recognizing our proper citations and our contribution in incorporating LLM priors into decision-making tasks. Our core motivation for Mem-EM with LLMs is to utilize a **compact decision-making memory** to refine the **LLM prior** for solving complex real-world decision-making tasks. Our Mem-EM provides a **seamless and compatible self-improvement framework** consisting of memory-based value estimation and LLM Prior refinement modules.
>
> **Algorithmic Novelty When LLMs Come In**
>
> While we are inspired by Episodic Control (EC) for effective non-parametric value estimation and adopt a probabilistic inference perspective (Control-as-Inference) for decision-making due to its strong formulation capability, **our work is not merely an application of existing EM-style policy updates with LLMs**.
>
> **The incorporation of LLM priors enables us to tackle complex real-world challenges**, such as the huge, non-enumerable action spaces found in environments like ALFWorld (up to 50 admissible actions per state), where traditional RL methods like DQN and pure EC fail to solve as shown in Fig 3.
>
> **Key Algorithmic Differences:**
>
> - **Mutual Reinforcement (Self-Improvement):** The overall framework establishes a closed loop where memory-driven Q-estimation guides the LLM policy's actions, and the high-quality experiences stored in memory are used to refine the LLM prior. This mutual reinforcement is central to our claim of a self-improvement framework.
>
> - **Action Space Handling and Efficiency:** Traditional control-as-inference methods like MPO rely on sampling from and estimating the action policy over the entire action space, **limiting their adaptability to complex scenarios** such as ALFWorld with huge and state-varying action spaces. In contrast, our method leverages the LLM prior to generating $K$ action candidates and then uses memory-driven Q-estimation to select the best action. This mechanism enables our method to adapt to complex unstructured action space settings. Furthermore, this  Q-value guided action sampling strategy effectively **narrows down the executable exploration space**, enabling efficient exploration.
>
> - **Modality and Generality**: Traditional EC operates over numerical states and small enumerable action spaces as separate modalities. Our Mem-Q exploits **the unified textual modality** in text-based MDPs, using LLM-based embeddings to jointly represent state-action pairs for Q-value estimation. This captures **more semantic similarity across actions** than EC and seamlessly handles diverse, unstructured textual action spaces, breaking through classic EC's limitations.

---

> ### Author Response · Authors · 2025-11-24
>
> ### **Weakness 2: Stability and Practicality Concerns**
> >Off-policy/Q-type-learning convergence is typically unstable. The paper even relies on importance sampling. Furthermore, the importance sampling weight is a heuristic approximation. Finally, does the memory table has sufficient coverage/exploratory enough? This makes people questions the practicality of the algorithms. I believe when we talk about policy optimization for LLMs, we have to be careful and understand what is new/different compared with standard DRL setting.
>
>
>
> We acknowledge the concerns regarding stability, particularly with off-policy methods and importance sampling. Our design prioritizes **algorithmic robustness and computational efficiency** specifically for the LLM setting:
>
> - **Stable Off-policy/Q-type-learning:** The Mem-Q module is inherently more stable than training a parametric Deep Q-Network (DQN) from scratch, as it performs non-parametric value estimation directly from stored, high-quality episodic returns ($R_t$). In addition, the incorporation of the LLM Prior combined with importance sampling of $K$ action candidates effectively **narrows the exploration action space**, thereby enabling **faster convergence** compared to traditional Q-learning methods.
>
> - **Heuristic for Efficiency:** The use of the memory distribution $q(s, a)$ to approximate the joint LLM prior $p_{LLM_{\theta}}(s, a)$ in the Self-Normalized Importance Sampling (SNIS) weight (Eq. 6) is a deliberate and effective heuristic choice. This approximation is empirically found to **stabilize training** and, crucially, **eliminates the need for repeated, expensive LLM queries** for prior estimation, making the LLM fine-tuning practical.
>
> - **Sufficient memory Coverage and Low Fine-Tuning Overhead**:
>
>   - **Memory Capacity:** The memory table employs a simple **Least-Recently-Used (LRU)** replacement strategy and proves robust. As shown in Fig. 4(c), on Overcooked, Mem-Q with only $N=200$ entries achieves performance comparable to $N=1000$, demonstrating that the stored representations retain **crucial information** efficiently. Thereby, thanks to the retrieval-based Q estimation and LLMs' embedding capacity, our method does not require storing every single state-action pair but efficiently leverages a compact subset of frequently-accessed experiences.
>
>   - **Prior Refinement Frequency:** Our method is practical because it requires **infrequent LLM fine-tuning**. For ALFWorld, we achieve high performance with only **six tuning events** throughout the training, demonstrating that the LLM prior maintains robustness and does not require constant updating, which is vital given the computational cost of LLM tuning.
>
> **Regarding the difference from standard DRL settings for policy optimization with LLMs**
>
> This work presents a seamless and compatible **memory-driven self-improvement framework** that integrates memory-based value estimation and LLM prior refinement modules. This architecture provides an efficient way for **incorporating compact, domain-specific knowledge into LLMs** to solve complex, real-world challenges.
>
> An intuitive understanding of its **difference from DRL on policy optimization** is as follows: traditional DRL methods for policy optimization, such as PPO, typically **sample a single action** for each state. In contrast, our memory-driven action policy performs **Q-value-guided search** by sampling $K$ candidates from the LLM prior and selecting the best action based on Q-value estimation. The ablation study on the inefficiency of sampling only $K=1$ action candidate, similar to the actor-critic RL setting (shown in Figure 4a) and the comparison with the policy-based RL baseline, PPO (detailed in response to Weakness 3 below), demonstrate that this mechanism is **crucial for efficient LLM policy optimization**.

---

> ### Author Response · Authors · 2025-11-24
>
> ### **Weakness 3: Limited Baselines**
>
> > Baselines studies are also limited. For example, no comparisons with search-based planners guided by value, no comparisons with more advanced memory-based RL algorithms.
>
>
> We have expanded our comparative analysis to address the reviewer's concern, specifically by including a direct comparison with a prominent policy-based LLM fine-tuning method and a memory-based LLM agent algorithm.
>
> - **Comparison with Policy-Based Fine-Tuning (PPO):** In the revised submission (Section 11.1), we compare ${Mem-EM}_{w/tune}$ against the **PPO** algorithm applied to LLM policy fine-tuning.
>
>   Results across Overcooked (Tomato), ALFWorld (Pick), and the sparse-reward and partially obsevale benchmark **VirtualHome** show that $Mem-EM_{w/tune}$ **significantly outperforms PPO**, thanks to our robust memory storage and value-guided action policy. This validates the superior **sample efficiency** of our EM-based memory-driven approach over standard on-policy LLM tuning.
>
>   **Comparison with Search-Based Planners:** While we did not include a general search-based planner (e.g., MCTS) as a baseline, our **Memory-Driven Action Policy** (Section 4.1) inherently performs a **Q-value-guided search** by sampling $K$ candidates from the LLM prior and selecting the action based on $\hat{Q}$. The ablation study shows this mechanism is crucial for performance (Figure 4a).
>
> - **Comparison with Advanced Memory-Based RL:** Our work is built upon and exceeds previous memory-based work:
>
>   - **Mem-Q** explicitly outperforms **DQN**, with memory for Q network learning  and classic **Episodic Control** by incorporating LLM semantic embeddings.
>   - We also compare against **DQN-Prior**, which represents an advanced baseline integrating a fixed LLM prior with a conventional Q-network. Our memory-based variants consistently achieve **comparable or superior performance**, confirming the benefits of memory-driven value estimation and LLM prior refinement.
>   - In addition, In the revised submission (Section 11.2), we compare our Mem-EM with a memory-based LLM agent, **Memorybank**[1]. Although both our  $\text{Mem-EM}_\text{w/tune}$ and MemoryBank involve memory for previous experiences and LLM priors, **our $\text{Mem-EM}_\text{w/tune}$ outperforms MemoryBank** on Overcooked. The key difference lies in how memory is utilized: our method uses memory for value estimation and LLM prior refinement combined with a value-guided action policy, whereas MemoryBank simply uses retrieved memory as additional decision context for the LLM' decision-making.
>
> - **Results on Overcooked(Tomato) with Qwen2.5-3B**
>
>   | Baselines/Training Episodes     | 50       | 100      | 150      | 200      |
>   | ------------------------------- | -------- | -------- | -------- | -------- |
>   | $\text{Mem-EM}_\text{w/tune}$   | **1.01** | **1.11** | **1.11** | **1.11** |
>   | $\text{Mem-EM}_\text{w/o tune}$ | 1.01     | 1.09     | 1.09     | 1.08     |
>   | *PPO*                           | 0.48     | 0.81     | 0.98     | 1.08     |
>   | *Memory-bank*                   | 0.40     | 0.51     | 0.52     | 0.65     |
>   | DQN-Prior                       | -0.03    | 0.03     | 0.81     | 1.13     |
>   | DQN                             | 0        | 0.07     | 0.09     | 0.29     |
>
> - **Results on ALFWorld(Pick) with Qwen2.5-3B**
>
>   | Baselines/Training Episodes     | 150      | 300      | 450      | 600      |
>   | ------------------------------- | -------- | -------- | -------- | -------- |
>   | $\text{Mem-EM}_\text{w/tune}$   | **0.40** | **0.78** | **0.82** | **0.93** |
>   | $\text{Mem-EM}_\text{w/o tune}$ | 0.32     | 0.30     | 0.38     | 0.47     |
>   | *PPO*                           | 0.18     | 0.38     | 0.43     | 0.58     |
>   | DQN-Prior                       | 0.03     | 0.08     | 0.38     | 0.35     |
>   | DQN                             | 0        | 0        | 0        | 0        |
>
> - **Results on VirtualHome with Qwen2.5-3B**
>
>   | Baselines/Training Episodes     | 50       | 100      | 150      | 200      |
>   | ------------------------------- | -------- | -------- | -------- | -------- |
>   | $\text{Mem-EM}_\text{w/tune}$   | **0.61** | 0.95     | 0.97     | **1.00** |
>   | $\text{Mem-EM}_\text{w/o tune}$ | 0.58     | **0.98** | **0.99** | 0.99     |
>   | *PPO*                           | 0.09     | 0.35     | 0.61     | 0.84     |
>   | DQN-Prior                       | 0.00     | 0.07     | 0.21     | 0.42     |
>   | DQN                             | 0        | 0        | 0        | 0        |
>   | Mem-Q                           | 0        | 0        | 0        | 0        |
>
>   [1]Zhong, Wanjun, et al. "Memorybank: Enhancing large language models with long-term memory." Proceedings of the AAAI Conference on Artificial Intelligence. Vol. 38. No. 17. 2024.

---

### Official Review · Reviewer_4VxH · 2025-11-01

**Soundness:** 3
**Presentation:** 3
**Contribution:** 2
**Rating:** 4
**Confidence:** 3

**Summary:**

This paper addresses the challenge of adapting Large Language Models (LLMs) for specific sequential decision-making (SDM) tasks, particularly when domain-specific data is limited. The authors note that while LLMs possess broad prior knowledge, this is often insufficient for specialized tasks. To solve this, the paper proposes a **memory-driven self-improvement framework**. The core of this framework is a memory table that stores past interactions (state-action pairs) and their estimated Q-values. This process forms a self-improvement loop: the refined LLM generates better trajectories, which in turn enriches the memory, leading to better value estimates and further refinement. The framework is formalized using an Expectation-Maximization (EM) algorithm. Experiments conducted on ALFWorld and Overcooked environments demonstrate that the proposed method (Mem-EM) significantly outperforms baselines like DQN and DQN-Prior in terms of sample efficiency and final performance, and shows strong generalization to unseen tasks.

**Strengths:**

- The paper addresses the **important and practical problem** of adapting generalist LLMs for specialized decision-making tasks, which is a key challenge for their real-world application.

- The core idea of using a **memory table to drive both value estimation and LLM prior refinement** is intuitive and well-motivated. The proposed self-improvement loop, where the memory and LLM mutually enhance each other, is a compelling concept.

- The paper is **well-written and easy to follow**. The framework's components and data flow are clearly explained, supported by effective illustrations.

**Weaknesses:**

The **empirical evaluation, while promising, could be strengthened** in several areas:

- The comparison is primarily against DQN and a DQN-Prior baseline. The exclusion of more advanced or policy-based RL algorithms (e.g., PPO, DAPO), which are common in decision-making, makes it difficult to fully contextualize the performance gains.

- The evaluation is limited to two benchmarks, ALFWorld and Overcooked. While these are relevant text-based environments, including **more complex decision-making tasks** would provide a more robust validation of the framework's scalability and effectiveness.

- The experiments rely exclusively on the Qwen model family (Qwen2.5-3B and 7B). Testing the framework with **other foundational LLMs** (e.g., from the Llama families) would be valuable to demonstrate that the approach is model-agnostic.

**Questions:**

- The paper mentions that the computational overhead is "tolerable" by using LORA and infrequent tuning (e.g., six times in the ALFWorld experiment). Could the authors provide a more concrete analysis of the **computational cost** (e.g., total training time, GPU memory) of the $Mem-EM_{w/tune}$ method compared to the baselines?

- For the memory-driven Q-estimation, the authors use an **inverse distance kernel** (Eq. 3). What was the rationale for this specific choice? Have other kernel functions or alternative non-parametric estimators been explored, and how do they compare?

---

> ### Author Response · Authors · 2025-11-23
>
> We thank the reviewer for their time and effort in reviewing our paper, we are glad that you appreciate 1)the **important and practical problem** , 2) the **intuitive and well-motivated self-improvement framework** and 3) the work being **well-written and easy to follow**. We have provided detailed explanations and clarifications to resolve your concerns and made several updates that have improved the paper based on your recommendation. We respectfully hope you can reconsider your score to reflect our updates.
>
> > Q1: Limited Baseline
>
> Thank you for your valuable suggestions. We have compared our method with the policy-based LLM-tuning approach **PPO** and a memory-based LLM agent algorithm**Memorybank**[1].
>
> Practically, in PPO baseline, the LLM acts as the action policy and is fine-tuned following the PPO algorithm. In the MemoryBank baseline, previous state-action pairs are stored in memory, and the LLM makes decisions based on retrieved state-relevant memory.
>
> As shown in the table below (see Figure 6 in Appendix for more details), although both methods involve fine-tuning the same LLMs, our **$\text{Mem-EM}_\text{w/tune}$ outperforms PPO** thanks to the memory storage and value-guided action policy as described in section 4.1.
>
> Although both our  $\text{Mem-EM}_\text{w/tune}$ and MemoryBank involve memory for previous experiences, **our $\text{Mem-EM}_\text{w/tune}$ outperforms MemoryBank**. The key difference lies in how memory is utilized: our method uses memory for value estimation and LLM prior refinement combined with a value-guided action policy, whereas MemoryBank simply uses retrieved memory as additional decision context for the LLM.
>
> **Results on Overcooked(Tomato) with Qwen2.5-3B**
>
> | Baselines/Training Episodes     | 50       | 100      | 150      | 200      |
> | -- | --- | -| - | -|
> | $\text{Mem-EM}_\text{w/tune}$   | **1.01** | **1.11** | **1.11** | **1.11** |
> | $\text{Mem-EM}_\text{w/o tune}$ | 1.01     | 1.09     | 1.09     | 1.08     |
> | PPO  | 0.48     | 0.81     | 0.98     | 1.08     |
> | Memory-bank| 0.40     | 0.51     | 0.52     | 0.65     |
> | DQN-Prior| -0.03    | 0.03     | 0.81     | 1.13     |
> | DQN   | 0        | 0.07     | 0.09     | 0.29     |
>
> **Results on ALFWorld(Pick) with Qwen2.5-3B**
>
> | Baselines/Training Episodes     | 150      | 300      | 450      | 600      |
> | ---| -------- | -------- | -------- | -------- |
> | $\text{Mem-EM}_\text{w/tune}$   | **0.40** | **0.78** | **0.82** | **0.93** |
> | $\text{Mem-EM}_\text{w/o tune}$ | 0.32     | 0.30     | 0.38     | 0.47     |
> | PPO| 0.18     | 0.38     | 0.43     | 0.58     |
> | DQN-Prior  | 0.03     | 0.08     | 0.38     | 0.35     |
> | DQN | 0        | 0        | 0        | 0        |
>
> [1]Zhong, Wanjun, et al. "Memorybank: Enhancing large language models with long-term memory." Proceedings of the AAAI Conference on Artificial Intelligence. Vol. 38. No. 17. 2024.
>
>
> > Q2: The evaluation is limited to two benchmarks
>
> To further verify the effectiveness of our framework, we have added experiments on an **additional text-based benchmark, VirtualHome[1]**, which is widely used to evaluate LLM-based decision-making [2] [3]. Following prior work, we consider the food preparation task, which requires the agent to complete the task within 4 rooms. This task is challenging due to **partial observability and sparse rewards**: only objects in the current room are visible, the agent receives reward 1 only upon task completion (0 otherwise), and the action space contains up to 10 possible actions.
>
> As shown in the table below (see Figure 7 in Appendix for details), our **Mem-EM significantly outperforms all other baselines**. Due to the sparse reward signal and partial observability, traditional RL algorithms, including DQN and Mem-Q, are unable to successfully explore. Methods that involve LLMs can incorporate rich prior knowledge into the exploration process, thus performing better than the LLM-free DQN and Mem-Q baselines. In addition, our **$\text{Mem-EM}_\text{w/tune}$ demonstrates superior sample efficiency and is more robust** than PPO and DQN-Prior.
>
> **Results on VirtualHome with Qwen2.5-3B**
>
> | Baselines/Training Episodes     | 50       | 100      | 150      | 200      |
> | - | -------- | -------- | -------- | -------- |
> | $\text{Mem-EM}_\text{w/tune}$   | **0.61** | 0.95     | 0.97 | **1.00** |
> | $\text{Mem-EM}_\text{w/o tune}$ | 0.58     | **0.98** | **0.99** | 0.99     |
> | PPO | 0.09     | 0.35     | 0.61     | 0.84     |
> | DQN-Prior| 0.00| 0.07     | 0.21     | 0.42     |
> | DQN| 0        | 0| 0 | 0   |
> | Mem-Q | 0| 0        | 0        | 0        |
>
>
> [1]Puig, Xavier, et al. "Virtualhome: Simulating household activities via programs." *CVPR*. 2018.
>
> [2] Wen, Muning, et al. "Reinforcing llm agents via policy optimization with action decomposition." *NeurIPS* 2024.
>
> [3] Tan, Weihao, et al. "True Knowledge Comes from Practice: Aligning Large Language Models with Embodied Environments via Reinforcement Learning." *ICLR* 2024.

---

> ### Author Response · Authors · 2025-11-23
>
> > Q3: The experiments rely exclusively on the Qwen model family
>
> Thank you for your valuable comments. To validate the effectiveness of our method across different foundation LLMs, **we added an experiment on fine-tuning three models: Qwen2.5-3B, Qwen2.5-7B, and LLaMA3-8B**, following our memory-driven LLM refinement. In practice, we use the memory table obtained by performing $\text{Mem-EM}_{\text{w/tune}}$ with Qwen2.5-7B on ALFWorld(Pick), collecting approximately 6,000 $(s,a,Q)$ memory tuples. The three LLMs are then fine-tuned using our proposed LLM prior refinement objective function (see Eq. 7) with this memory table. As shown in the table below, all **there LLMs achieve significant performance gains with our LLM fine-tuning objectives**, demonstrating that the decision-making knowledge reflected in the memory table  can be successfully distilled into both the smaller model Qwen2.5-3B for more effective deployment and a model with a different architecture, LLaMA3-8B.
>
> **Results of fine-tuning different LLMs on ALFWorld(Pick)**. All models are fine-tuned with the same hyperparameters: learning rate of 5e-4, LoRA rank of 16, and 5 epochs.
>
> |            | Without Finetune | Finetuned |
> | ---------- | ---------------- | --------- |
> | Qwen2.5-3B | 0.05             | 0.85      |
> | Qwen2.5-7B | 0.10             | 0.88      |
> | LLaMA3-8B  | 0                | 0.83      |
>
> > Q4: About Computation Cost
>
> Thank you for your valuable suggestions. We have added detailed computation costs of the baselines on ALFWorld(Pick) with Qwen2.5-3B, as shown in the table below. Our method achieves significant performance gains over PPO and DQN-Prior while maintaining similar training time and GPU usage requirements as the previous best-performing method, PPO.
>
> |                                          | PPO     | DQN-Prior | $\text{Mem-EM}_{\text{w/tune}}$ | $\text{Mem-EM}_{\text{w/o tune}}$ |
> | ---------------------------------------- | ------- | --------- | ------------------------------- | --------------------------------- |
> | Performance                              | 0.62    | 0.38      | 0.93                            | 0.55                              |
> | Training Time                            | 11.05h  | 12.07h    | 11.20h                          | 13.57h                            |
> | Training GPU Usage                       | 13.34GB | 8.78GB    | 15.45GB                         | 7.8GB                             |
> | Memory CPU Usage                         | /       | /         | 57.42MB                         | 61.12MB                           |
> | Memory Table Size                        | /       | /         | 7100                            | 7554                              |
> | Inference time(query 1000 times)/minutes | 3.40    | 3.59      | 5.28                            | 8.16                              |
>
> > Q5: “For the memory-driven Q-estimation, the authors use an **inverse distance kernel** (Eq. 3). What was the rationale for this specific choice? Have other kernel functions or alternative non-parametric estimators been explored, and how do they compare？”
>
> We use the Inverse Distance Kernel (Eq. 3) for non-parametric Q-value estimation, inspired by its proven success in previous weighted estimation methods (such as NEC[1]). This choice enables **strong locality** that the Q-estimate relies heavily on the nearest and most relevant past experiences. As mentioned in NEC, The **Gaussian Kernel would exponentially suppress all neighbors** except for the closest one. In contrast, the kernel we chose has the advantage of having heavy tails and does not require extensive kernel hyperparameter tuning. **Cosine Similarity** was unsuitable as it focuses on vector direction but **ignores vector magnitude**, violating the strict locality requirement.
>
> [1]Pritzel, Alexander, et al. "Neural episodic control." *International conference on machine learning*. PMLR, 2017.

---

### Author Response · Authors · 2025-11-24
**Summary of Revision**

We sincerely thank all reviewers for their favour of our work and deeply appreciate the time and thoughtful feedback provided, as it has significantly contributed to improving our work. Now the revised version is ready. We summarise the main revision in the following:



- As the reviewer **4VxH**, **dA4m** and **po9h** suggested, we have included a comparison with policy-based LLM fine-tuning using PPO and a memory-based LLM agent MemoryBank, in Appendix 11.1.
- As the reviewer **4VxH**, **AnoS** and **po9h** suggested, we have included an additional benchmark, VirtualHome, which presents partial observability and sparse reward challenges, as shown in Appendix 11.2.
- As the reviewer **4VxH** suggested, we have included testing effectiveness of our framework across different foundation LLMs in Appendix 11.3.
- As the reviewer **4VxH**, **PotS** and **po9h** suggested, we have included the computational resource and time requirements during training and inference in Appendix 11.4.
- As the reviewer **PotS** and **po9h** suggested, we have included ablation studies on the exploration-exploitation trade-off parameter (**PotS**) and reward discount factor (**po9h**) in Appendix 11.5.
- As reviewer **dA4m** suggested, we have included a discussion of the differences from traditional episodic control and control-as-inference introduced by incorporating LLMs, highlighted in Sections 3 and 4.2.

Once again, we sincerely appreciate the constructive feedback you’ve provided. We would be glad to hear any further suggestions you might have that could help improve our work.

---

### Note · Authors · 2026-01-06

I have read and agree with the venue's withdrawal policy on behalf of myself and my co-authors.